# CycleResearcher: Improving Automated Research via Automated Review

WARNING: This work is not advocating the use of LLMs for paper writing.

**Yixuan Weng**[1,2*]**, Minjun Zhu**[2,3]*****Guangsheng Bao**[2,3]**, Hongbo Zhang**[2,3]**, Jindong Wang**[4]
**Yue Zhang**[1,2†]**, Linyi Yang**[5†]
[1]Research Center for Industries of the Future, Westlake University
[2]School of Engineering, Westlake University [3]Zhejiang University [4] William&Mary
[5]University College London
wengsyx@gmail.com, yanglinyiucd@gmail.com
zhuminjun,baoguangsheng,zhanghongbo,zhangyue@westlake.edu.cn

## Abstract

The automation of scientific discovery has been a long-standing goal within the research community, driven by the potential to accelerate knowledge creation. While significant progress has been made using commercial large language models (LLMs) as research assistants or idea generators, the possibility of automating the entire research process with open-source LLMs remains largely unexplored. This paper explores the feasibility of using open-source post-trained LLMs as autonomous agents capable of performing the full cycle of automated research and review, from literature review and manuscript preparation to peer review and paper refinement. Our iterative preference training framework consists of CycleResearcher, which conducts research tasks, and CycleReviewer, which simulates the peer review process, providing iterative feedback via reinforcement learning. To train these models, we develop two new datasets, Review-5k and Research-14k, reflecting real-world machine learning research and peer review dynamics. Our results demonstrate that CycleReviewer achieves promising performance with a 26.89% reduction in mean absolute error (MAE) compared to individual human reviewers in predicting paper scores, indicating the potential of LLMs to effectively assist expert-level research evaluation. In research, the papers generated by the CycleResearcher model achieved a score of 5.36 in simulated peer reviews, showing some competitiveness in terms of simulated review scores compared to the preprint level of 5.24 from human experts, while still having room for improvement compared to the accepted paper level of 5.69. This work represents a significant step toward fully automated scientific inquiry, providing ethical safeguards and exploring AI-driven research capabilities. The code, dataset and model weight are released at https://wengsyx.github.io/Researcher/.

## 1 Introduction

Automating general scientific discovery has been a long-standing ambition of the research community, dating back to the late 1970s and 1980s (Lenat, 1977; 1983; Langley, 1987) with the advent of computer science. In the field of AI, researchers have envisioned automating scientific research using AI itself (Hutter, 2001; Radensky et al., 2024). The recent emergence of large language models (LLMs) has opened new possibilities for this endeavor (Wang et al., 2023a; Lu et al., 2024), demonstrating their capacity to not only process but also contribute meaningfully to scientific research. Most current efforts have relied on commercial LLMs to build agents that propose research ideas (Wang et al., 2023b; Yang et al., 2023; Radensky et al., 2024; Baek et al., 2024; Liu et al., 2024b), as an assistant to conduct experiments (Du et al., 2024; Yang et al., 2024b; Li et al., 2024), or act as

---

*Equal Contribution
†Corresponding Authors. Supported by Research Center for Industries of the Future, Westlake University.

an AI scientist capable of generating automated open-ended scientific publications (Lu et al., 2024; Taniguchi et al., 2024). To date, the challenge of automating the entire scientific discovery process remains largely unresolved, particularly when it comes to generating and refining research outputs that meet the high standards of peer-reviewed work. Moreover, few efforts address the integration of iterative feedback, which is essential for maintaining academic soundness and novelty. Current models often struggle to adapt across the full spectrum of research stages, highlighting gaps in their ability to conduct comprehensive, multi-step scientific discovery.

Central to the scientific process is the iterative cycle of submission, peer review, and refinement – an established mechanism that maintains the quality and integrity of academic work (Smith, 2006; Boughton et al., 2018). Feedback from reviewers and peers plays a critical role in this cycle, offering insights that help researchers refine their work and improve its rigor and impact. Drawing inspiration from this cyclical process, we propose a novel framework that post-trains LLMs as autonomous agents to simulate the full loop of the scientific discovery process. Our approach, built entirely on open-source models, aims to replicate the real-world dynamic of research development and peer review processes. By leveraging trainable models, we enable the utilization of the iterative preference training mechanism (Yuan et al., 2024) using sampling examples through reinforcement learning. Our objective is to determine whether LLMs can actively contribute to each stage of scientific inquiry, from literature review and idea generation to experimental design, manuscript preparation, peer review and paper refinement.

Automating the entire research lifecycle, "Oberg et al. (2022) presents a significant challenge to current agent-based methods (Lu et al., 2024; Si et al., 2024; Yang et al., 2024b), which predominantly rely on commercial models. Consequently, these methods cannot be effectively modeled as policy optimization problems using reinforcement learning. While self-correction methods (Weng et al., 2023; Yuan et al., 2024; Lee et al., 2024) have been developed to enhance reasoning performance by assessing the quality of LLM outcomes, they have not yet been adopted in the domain of paper writing, which demands more complex evaluations from multiple perspectives. Our research addresses this gap by introducing an iterative post-training framework. The central research question we pose is: "*How can we automate the Research-Review-Refinement process by post-training LLMs?*" So that automated research can be improved according to feedback from automated reviews.

We build a novel iterative training framework (Pang et al., 2024) that contains two core components: the policy model (namely CycleResearcher) and the reward model (namely CycleReviewer) ranging in size – from 12B to 123B – based on Mistral (Jiang et al., 2023) and Qwen 2.5 (Yang et al., 2024a; Team, 2024).In our framework, CycleResearcher acts as a scientific thinker, responsible for reading literature, identifying research problems, proposing solutions, and designing experiments, while specific experiment execution is delegated to specialized code models. In particular, the policy model performs a variety of research tasks ... for paper generation[1]. The reward model, on the other hand, simulates the peer review process, evaluating the quality of the research output and providing feedback that informs reinforcement learning rewards. In the virtual RL environment, to accelerate training, we require the *experimental results* to be fabricated instead of conducting actual experiments

To illustrate our framework's operation, when exploring the topic of "Hacking Rewards of VLMs," we first fed fine-tuned CycleResearcher with a set of relevant published papers to inspire it to propose novel ideas. After generating a batch of first-round papers corresponding with those ideas, fine-tuned CycleReviewer evaluates them to generate pairwise preference samples, which are used to optimize the policy model using SimPO (Meng et al., 2024), this process is repeated.

For training our models, we construct two large-scale, publicly available datasets: Review-5k and Research-14k (described in Section 2), which contain peer review and accepted papers from major ML conferences (e.g., ICLR, ICML, NeurIPS). For testing, we take both subjective human evaluation and objective model-based evaluations to assess the quality of CycleReviewer and CycleResearcher. Our experiments show that CycleReviewer demonstrates promising capabilities in supporting the peer review process, while CycleResearcher exhibits consistent performance in research ideation and experimental design compared to API-based agents (Lu et al., 2024). We also acknowledge that the generalizability across research domains remains a challenge for current LLMs. Our contributions can be summarized as:

---

[1]Our current implementation delegates experiment execution to code generation models, allowing CycleResearcher to focus on high-level research planning and analysis. Notably, the *experimental results* generated by CycleResearcher in this work are fabricated and do not represent real experimental data.

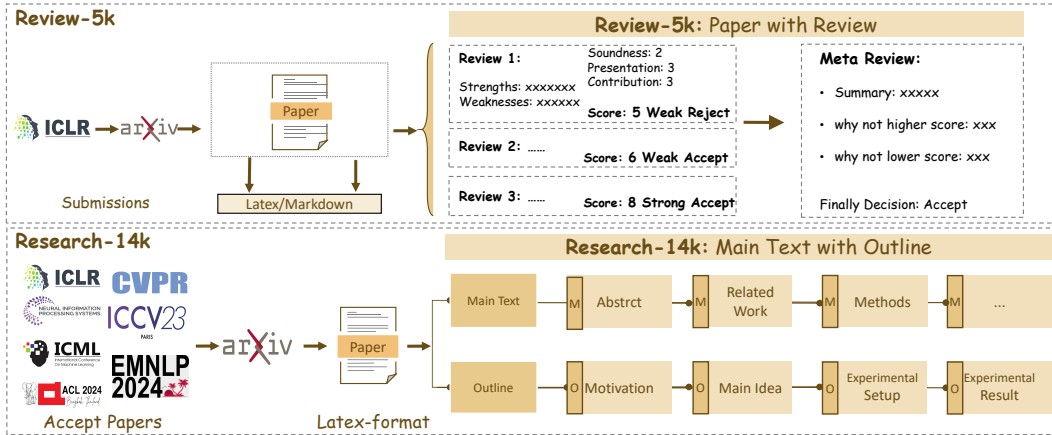

Figure 1: Data Construction pipeline of the Research-14k dataset and Review-5k dataset. The Research-14k dataset includes both the **main text (M)** and **outlines (O)** of research papers, covering key components such as motivation, methods, experimental setup, and results. The Review-5k dataset provides 3 reviews and 1 meta-review for each paper

- We introduce an iterative reinforcement learning framework that automates the entire research lifecycle, which mirrors the real-world Research-Review-Refinement cycle. Our framework includes *CycleResearcher*, a policy model for research tasks, and *CycleReviewer*, a reward model simulating peer reviews. This framework enables large language models (LLMs) to iteratively improve research outputs through a Research-Review-Refinement cycle.

- We release two large-scale datasets, *Review-5k* and *Research-14k*, which are publicly available and designed to capture the complexity of both peer review and research paper generation in machine learning. These datasets provide valuable resources for evaluating and training models in academic paper generation and review.

- We demonstrate that the CycleResearcher model can generate papers with an average quality level close to human-written preprints, achieving an acceptance rate of 31.07%. Furthermore, our *CycleReviewer* model shows encouraging results with a 26.89% improvement in MAE compared to individual reviewers, suggesting the potential of automated research assessment in mean absolute error (MAE) in research evaluation tasks, setting a new benchmark for automated research assessment.

## 2 DATASET CONSTRUCTION

In this section, We present an overview of how we collect a substantial corpus of academic papers and organize them into the **Review-5k** and **Research-14k** training dataset. As illustrated in Figure 1, we introduce structured outline extraction and segmentation to assist the LLM in planning before generating research papers. **Notably, we will only make our datasets publicly available for those papers, which receive written consent from publishers (See in Appendix C).**

### 2.1 REVIEW-5K

In order to collect a high-quality review dataset, we first gather paper information (including title, abstract, and PDF data) along with the corresponding review comments from ICLR 2024. This ensures that all papers are evaluated according to a consistent standard. We then attempt to retrieve the permitted LaTeX files from ArXiv. If the LaTeX files are unavailable, we use MagicDoc to convert the retrieved PDFs into markdown format. Then, inspired by the traditional peer review process, where a group of reviewers evaluates a paper, followed by a senior reviewer who synthesizes their feedback and makes the final decision, we collect each data point including key components: 1) summary of the work, 2) identified strengths and weaknesses, and 3) questions for clarification, along with 4) numerical scores for soundness, presentation, contribution, and an overall rating. Finally, we leave a dataset named Review-5k, containing 4,991 papers collected from ICLR 2024, comprising over 16,000 reviewer comments. Finally, we split our dataset into mutually exclusive training/testing sets, we keep 4,189 paper reviews for training and 782 samples for testing.

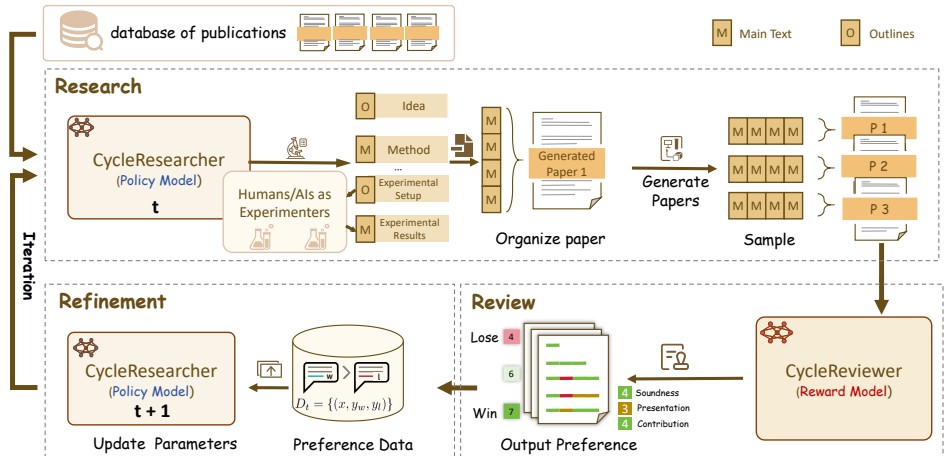

Figure 2: **Iterative Training Framework**. The CycleResearcher model generates Outline (O) and main texts (M) to organize papers, which are evaluated by the CycleReviewer and constructed into preference pairs based on rewards. This whole procedure is then iteratively refined, resulting in progressively enhanced research abilities with each iteration.

## 2.2 RESEARCH-14K

The research-14k dataset aims to capture structured outlines and detailed main text from academic papers. The data construction process involves three steps: **(1)**. we first compile a list of accepted papers from major international machine learning conferences, such as ICLR, NeurIPS, ICML, ACL, EMNLP, CVPR, and ICCV, spanning from 2022 to 2024. Using Semantic Scholar[2], we retrieve the corresponding ArXiv links and LaTeX-format files for each paper, gathering a total of 14,911 papers. The main text of these papers is then pre-processed using rule-based filtering to remove irrelevant content such as comments ("%") and acknowledgments. **(2)**. Since the academic value of a research paper depends on its background, we also use the Semantic Scholar API to retrieve the cited works from the bib file and add their abstracts to it. **(3)**. Finally, we organize the main body of each paper into outlines and separate sections to help the model better understand the research process. We use the Mistral-Large-2 model (Jiang et al., 2023) to extract outline information from the paper, following the outline structure shown in Figure 1, and concatenate each outline with its corresponding section. These components form the complete fine-tuning dataset, where the input consists of detailed reference files and the output contains the paper outlines and main text.

After filtering papers that do not meet the requirements, the final dataset, Research-14k, includes 12,696 training samples and 802 test samples. It covers nearly all significant machine learning papers from the past three years and ensures that all collected papers are open-access. The training and test sets are split chronologically, with test papers published later than the training ones. This dataset is used for supervised fine-tuning to enable the LLM to generate well-structured academic papers. Additionally, Research-14k is a long-output dataset, with an average output length of 28K tokens.

## 3 ITERATIVE TRAINING FRAMEWORK

We use the iterative Simple Preference Optimization (SimPO) (Meng et al., 2024) framework to replicate the Research-Review-Refinement cycle typical in academic research. As mentioned in the introduction, we primarily focus on the development of ideas and the writing process, while the execution of actual experiments (Liu et al., 2024b; Zhu & Zhou, 2024; Hu et al., 2025) is beyond the scope of this work. The process begins with initializing two models: a baseline language model fine-tuned for academic writing (the CycleResearcher), and an LLM specialized in evaluating research papers (the CycleReviewer).

As illustrated in Figure 2, each iteration encompasses two primary phases: (1) The CycleResearcher model simulates key research steps, including literature review, hypothesis formulation, experimental

---

[2]https://www.semanticscholar.org/

design, and paper writing, culminating in the production of an academic paper. (2) Subsequently, the CycleReviewer model simulates a peer review process based on the generated paper, providing comprehensive feedback and quantitative scores. To facilitate iterative improvement, we implement a resampling procedure after each round, generating new preference data based on paper scores, which is then utilized to train the models for the subsequent iteration.

## 3.1 REWARD MODEL: CYCLEREVIEWER

We train CycleReviewer as the Generative Reward model on the Review-5k Dataset. To accurately reflect the academic peer review process, we establish a streamlined evaluation workflow:

$$\text{Paper} \rightarrow R_1, R_2, \ldots, R_n \rightarrow \text{SR}, \tag{1}$$

Where the research paper (Paper) is reviewed by multiple reviewers $(R_1, R_2, \ldots, R_n)$. Each reviewer's opinion is then summarized by a Senior Reviewer (SR), forming the final decision.

The input to the *CycleReviewer* model is a complete research paper. Upon receiving the paper, the model generates sequential feedback and scores for key aspects including Strengths, Weaknesses, Soundness, Presentation, Contribution, and an Overall Score. The Overall Score is rated on a scale from 1 to 10, where 1 represents the lowest score and 10 the highest, with 5 indicating the paper is borderline for rejection and 6 suggesting it is near acceptance. The output of the model includes both the Overall Score and a recommendation labelled as the "Final Suggestion." The CycleReviewer simulates the review process across multiple reviewers, producing a set of Overall Scores. The final output is the average of these scores, representing the overall evaluation of the paper by the system.

**Settings.** We use the Mistral-Large-2 model with LoRA-GA (Wang et al., 2024a) on an 8x H100 80G cluster, with a learning rate of 1e-5 and a batch size of 4x8, for 12 epochs on the Reviewer-5k dataset. To ensure diversity in the generated reviews, *CycleReviewer* starts by simulating the feedback from the reviewer with the lowest rating, gradually progressing to the highest-rated reviewer. This approach ensures that a range of perspectives, from more critical to more favorable, are considered before the senior reviewer delivers the final assessment.

## 3.2 POLICY MODEL: CYCLERESEARCHER

The CycleResearcher model is trained on Research-14k, and the process begins with a literature review, where the input bib file contains all references and their corresponding abstracts. After gaining a comprehensive understanding of the research background, the model moves on to manuscript preparation. In this stage, generating outlines and main text alternates to ensure a logical flow. First, the model generates the motivations and main ideas in the outline and then follows up by producing the title, abstract, introduction, and method sections in the main text. Next, it outlines the experimental setup and results, and subsequently generates the experimental design and simulated results in the main text, where it also incorporates discussions. In the virtual RL environment, to accelerate training, we require the "experimental results" to be fabricated instead of conducting actual experiments. Finally, the model analyzes the experimental results and formulates the conclusion. Once all sections of the main text are generated, they are combined into a complete paper in LaTeX format. Notably, each part of the research paper in Research-14k is precisely segmented. Finally, the generated paper $P$ is evaluated using the CycleReviewer, as described in Section 3.1.

**Settings.** To build the policy model, we select widely used open-source LLMs: Mistral-Nemo-12B, Qwen2.5-Instruct-72B, and Mistral-Large-2 123B. All models are trained using 8x H100 GPUs and DeepSpeed + ZeRO2 (Rajbhandari et al., 2020; Rasley et al., 2020). We maximized context length by setting the 12B model to 32K tokens, while the 72B and 123B models were set to 24K tokens. Given memory constraints, samples exceeding the preset context length are randomly truncated. We use a batch size of $2 \times 8$, a learning rate of $4e^{-5}$, and train for a total of 12,000 steps. These models support context windows up to 128K tokens, making them suitable for planning research projects and writing research papers. In response, we contribute three versions of policy models: CycleResearcher-12B, CycleResearcher-72B, and CycleResearcher-123B. During the reinforcement learning phase, we used a learning rate of 5e-7. For the 12B model, we used a text length of 18K, while for the 72B and 123B models, the maximum text length was 10K, with truncation applied from the end. Each iteration trains for one epoch using data obtained through sampling.

## 3.3 ITERATIVE SIMPO

We design an Iterative preference optimization alignment method (Xiong et al., 2024; Liu et al., 2024a) that simulates the peer review process as a reward mechanism. To construct a preference-pair dataset, we first collected 4,152 recent machine learning papers published on arXiv, retaining only the reference sections as the knowledge base. Then we sampled three times from the CycleResearcher with a temperature of 0.4 and processed the results into standard LaTeX-style texts $M_1, M_2, M_3$. Next, the CycleReviewer model simulated discussions among multiple reviewers, providing detailed evaluations of various aspects of the papers (e.g., novelty, methods, experimental design, result analysis). The average score $r_i$ from all simulated reviewers was assigned to each output $M_i$. We then selected the output with the highest reward value as the positive sample $y_w$ and the one with the lowest reward value as the negative sample $y_l$, forming a preference-pair dataset $D_0 = (x, y_w, y_l)$.

**Policy Optimization.** Instead of using the iterative DPO training framework (Pang et al., 2024), we adopt the SimPO as the base method for saving computational costs. To mitigate overfitting, we sample one-third of the full dataset in each round. Then, we generate a series of models $P_1, \ldots, P_T$, where each model $P_{t+1}$ is created using the preference data $D_t$ generated from the model $P_t$. With the preference-pair dataset, we trained a new policy model $\pi_\theta$ from $P_t$ to $P_{t+1}$. $P_1$ was initialized from the original fine-tuned CycleResearcher model using instruction tuning.

SimPO builds upon DPO (Rafailov et al., 2023), which is one of the most common offline preference optimization methods. It introduces a length-normalized reward function aligned with the generation target, thereby eliminating dependence on a reference model $\pi_{\text{ref}}$, which reduces memory and computation requirements. The reward function for SimPO is as follows:

$$r_{\text{Simpo}}(x, y) = \frac{\beta}{|y|} \log \pi_\theta(y \mid x) = \frac{\beta}{|y|} \sum_{i=1}^{|y|} \log \pi_\theta(y_i \mid x, y_{<i}), \tag{2}$$

where $\pi_\theta$ is the policy model, $|y|$ represents the length of the generated sequence, and $\beta$ is a constant controlling the scaling of reward differences. SimPO also introduces a target reward margin $\gamma > 0$ to help differentiate between winning and losing responses. The objective for SimPO is as follows:

$$\mathcal{L}_{\text{SimPO}}(\pi_\theta) = -\mathbb{E}_{(x, y_w, y_l) \sim \mathcal{D}} \left[ \log \sigma \left( \frac{\beta}{|y_w|} \log \pi_\theta(y_w|x) - \frac{\beta}{|y_l|} \log \pi_\theta(y_l|x) - \gamma \right) \right] \tag{3}$$

Considering that the models used in the research process may involve complex reasoning and mathematical calculations, we combine the SimPO loss learned from preference pairs with the negative log-likelihood (NLL) loss to stabilize training (Pang et al., 2024). The loss function for each preference pair is as follows:

$$\mathcal{L}_{\text{Our}}(\pi_\theta) = -\mathbb{E}_{(x, y_w, y_l) \sim \mathcal{D}} \left[ \log \sigma \left( \frac{\beta}{|y_w|} \log \pi_\theta(y_w|x) - \frac{\beta}{|y_l|} \log \pi_\theta(y_l|x) - \gamma \right) \right]$$
$$- \lambda \mathbb{E}_{(x, y_w) \sim \mathcal{D}_{\text{NLL}}} \left[ \log \pi_\theta(y_w \mid x) \right]. \tag{4}$$

Here, the hyperparameter $\lambda$ balances the two loss terms. Each round of training resamples and optimizes based on the previous round's results, enabling an approximate online policy optimization process, which allows the CycleResearcher to continuously adapt to evolving publication standards.

## 3.4 SAFEGUARD ACADEMIC INTEGRITY

Beyond automating the research process, we are also concerned with safeguarding academic integrity. We aim to prevent the misuse of LLMs in the research community. To achieve that, we adopt the Fast-DetectGPT (Bao et al., 2024), which aims to use the metric of conditional probability curvature to determine whether the paper submission is generated by LLMs. Specifically, we use Llama-3-8B (Dubey et al., 2024) as the scoring model and determine if a paper was generated by an LLM by comparing the conditional probability curvature with a predefined threshold $\epsilon$. If the curvature of a paper is larger than the threshold, we classify the paper as LLM-generated, otherwise human-written.

## 4 EXPERIMENTS

### 4.1 EXPERIMENTS ON PAPER REVIEW GENERATION

**Evaluation Metrics.** Evaluating reviewer performance is inherently difficult because the true quality of submissions is unknown. To address this challenge, we use Proxy Mean Squared Error (Proxy MSE) and Proxy Mean Absolute Error (Proxy MAE) to assess the accuracy of individual review scores (Su et al., 2024), detailed in Appendix E. For each paper, the conventional MSE and MAE for a review score $r$ are defined as $E\left[(r - \text{ground truth})^2\right]$ and $E\left[|r - \text{ground truth}|\right]$, which are unobservable due to the unknown true quality. Therefore, we introduce a proxy evaluation method using an independent, unbiased estimator as a stand-in for the ground truth score. Assuming we have $n$ human experts with scores $R = r_1, r_2, \ldots, r_n$, we treat each reviewer's score $r_i$ as an unbiased estimator of the true quality. We define $r_i' = \text{mean}(R \setminus r_i)$, which serves as an unbiased estimator excluding $r_i$. Thus, we measure the quality of $r_i$ using Proxy MSE $= (r_i - r_i')^2$ and Proxy MAE $= |r_i - r_i'|$. Simply put, for each submission, we use the average of the other $n - 1$ reviewers' scores as an estimator of the true score.

Our evaluation on the Reviewer-5k test set (average rating 5.53) uses this proxy approach for fair comparison. In the $n - 1$ mode, we randomly select one reviewer and use the average of remaining scores as the proxy ground truth. We apply this methodology to evaluate both human experts and closed-source models, including the AI Scientist review system (Lu et al., 2024) with one-shot reviews, self-reflection (Shinn et al., 2023), and ensembled reviews.

**CycleReviewer introduces better quality of review.** Table 1 presents the performance comparison across various models. CycleReviewer demonstrates encouraging results in peer review tasks compared to both proprietary systems and individual human reviewers. Our model shows a

Table 1: Comparison of automated models on generating review.

| Method | Proxy (Reviewer=$n-1$) MSE ↓ | MAE ↓ | Proxy (Reviewer=$n$) MSE ↓ | MAE ↓ | Decision Accuracy ↑ | Macro F1 ↑ |
|---|---|---|---|---|---|---|
| **Expert Individual** | 2.34 | 1.16 | - | - | **75.40%** | **75.39** |
| GPT-4o-mini | 3.44 | 1.53 | 2.98 | 1.40 | 53.06% | 34.72 |
| GLM-4 | 4.45 | 1.81 | 3.91 | 1.70 | 49.49% | 33.10 |
| DeepSpeek-2.5 | 4.62 | 1.83 | 3.72 | 1.64 | 45.11% | 39.98 |
| Gemini-1.5-pro | 3.02 | 1.34 | 2.56 | 1.23 | 50.98% | 50.75 |
| Claude-3.5-Sonnet | 6.40 | 2.23 | 5.62 | 2.12 | 48.05% | 32.44 |
| GPT-4o | 6.61 | 2.24 | 6.53 | 2.30 | 52.58% | 34.51 |
| **CycleReviewer (123B)** | **1.43** | **0.92** | **1.25** | **0.87** | 74.24% | 73.99 |

48.77% reduction in Proxy MSE and a 26.89% reduction in Proxy MAE when compared to individual reviewers' scores. These metrics suggest that CycleReviewer can provide consistent scoring that complements human expertise. With a decision accuracy of 74.24%, the model demonstrates competitive performance compared to other closed-source systems. These results suggest that our model can provide consistent scoring that complements human expertise, showing potential advantages over AI Scientist systems (Lu et al., 2024) in generating reliable evaluation scores. However, we emphasize that these metrics focus on score consistency rather than capturing the full complexity of expert review, where human insight remains invaluable.

### 4.2 THE IMPORTANCE OF RESEARCH LIFECYCLE SIMULATION

Table 2: The evaluation results of a series of papers assessed by CycleReviewer. The range of these scores is 1-10. The CycleReviewer simulates a group of reviewers, and we report the average score for the lowest Overall Score, the average score for the highest Overall Score, and the overall average score. † indicates that all these papers were actually accepted for publication.

| Paper Type | Source | Overall Score Metrics Avg Min Score ↑ | Avg Max Score ↑ | Avg Score ↑ | Accept Rate |
|---|---|---|---|---|---|
| Conference Accept Papers | Human Expert | **3.91** | **6.98** | **5.69** | **100.00%** † |
| Preprint Papers | Human Expert | 3.24 | 6.62 | 5.24 | 29.63% |
| AI Scientist | AI | 2.20 | 5.70 | 4.31 | *0.00%* |
| **CycleResearcher-12B (Ours)** | AI | 3.47 | **6.75** | 5.36 | **35.13%** |
| **CycleResearcher-72B (Ours)** | AI | **3.65** | 6.58 | **5.38** | 33.64% |
| **CycleResearcher-123B (Ours)** | AI | 3.30 | 6.45 | 5.15 | 24.28% |

Table 2 presents the results of CycleResearcher, which simulates a program committee review process, evaluating papers across the entire score range and ultimately providing a final acceptance decision

Table 3: The evaluation results of papers reviewed by CycleReviewer across three criteria: Soundness, Presentation, and Contribution. The range of these scores is 1-4.

| Paper Type | Source | Soundness Score | | | Presentation Score | | | Contribution Score | | |
|---|---|---|---|---|---|---|---|---|---|---|
| | | Min. ↑ | Max. ↑ | Avg. ↑ | Min. ↑ | Max. ↑ | Avg. ↑ | Min. ↑ | Max. ↑ | Avg. ↑ |
| Conference Accept Papers | Human Expert | **2.03** | **3.21** | **2.83** | **2.24** | **3.35** | **2.91** | **1.94** | **3.17** | **2.72** |
| Preprint Papers | Human Expert | 1.76 | 3.16 | 2.70 | 2.07 | 3.28 | 2.80 | 1.75 | **3.13** | 2.57 |
| AI Scientist | AI | 1.20 | 3.10 | 2.48 | 1.70 | **3.40** | 2.69 | 1.30 | 2.90 | 2.15 |
| **CycleResearcher-12B (Ours)** | AI | 1.73 | **3.17** | **2.71** | 1.91 | 3.24 | 2.70 | 1.68 | 3.07 | **2.60** |
| **CycleResearcher-72B (Ours)** | AI | **1.86** | 3.13 | **2.71** | **2.19** | 3.31 | **2.88** | **1.81** | 3.04 | 2.55 |
| **CycleResearcher-123B (Ours)** | AI | 1.74 | 3.14 | 2.69 | 2.10 | 3.31 | 2.83 | 1.72 | 3.08 | 2.53 |

based on simulated reviews. We report the average scores for the lowest-scoring reviewer, the highest-scoring reviewer, and the overall score. For accepted papers, we use the test set of Research-14k, where all papers have been accepted, serving as a benchmark for human expert standards. For preprint papers, we evaluate 955 submissions from arXiv (Sep. 2024) in the domains of cs.ML, cs.CV, and cs.LG. Additionally, we evaluate the AI Scientist with a collection of 10 research papers generated by GPT-4o and Claude-3.5.

**CycleResearcher consistently performs better than AI Scientist in terms of automated review metrics.** Table 2 shows that CycleResearcher-12B achieves an average score of 5.36, approaching the 5.69 average scores for conference-accepted papers and surpassing AI Scientist's score of 4.31. Notably, it achieves an acceptance rate of 35.13%, which is significantly higher than AI Scientist's 0% acceptance rate, demonstrating its superior ability to produce research-quality output.

The comparison across soundness, presentation, and contribution further illustrates the advantages of CycleResearcher. The CycleResearcher-12B achieves an average soundness score of 2.71, surpassing AI Scientist (with GPT-4o)'s 2.48 and closely matching the 2.83 of accepted papers. For presentation and contribution, it attains average scores of 2.70 and 2.60 respectively, outperforming AI Scientist's scores in both metrics (2.69 and 2.15). In contrast, AI Scientist shows significant limitations, particularly in minimum scores for soundness (1.20) and contribution (1.30), indicating less consistency in producing quality research. Our model demonstrates greater reliability, with higher minimum scores across all metrics (soundness: 1.73, presentation: 1.91, contribution: 1.68) compared to AI Scientist. These results underscore our model's effectiveness in addressing the challenges of ensuring quality and consistency in AI-generated research.

**Rejection sampling improves the quality of generated papers in terms of automated review metrics** in Figure 3. Rejection sampling is especially valuable in the context of academic paper generation, where the cost of producing research plans and papers using language models is relatively low compared to other stages of research. As the number of generated papers increases from 1 to 100, the average score rises from approximately 5.36 to 7.02, surpassing both preprint papers

Table 4: Ablation study of different variations of CycleResearcher-12B.

| Method | Avg Score ↑ | Accept Rate ↑ |
|---|---|---|
| **CycleResearcher** | **5.36** | **35.14%** |
| **w/o RL** | (−0.24)5.12 | (−5.34%)29.80% |
| **w/o Iterative** | (−0.15)5.21 | (−2.23%)32.91% |
| **w/o NLL** | (−0.45)4.91 | (−23.11%)12.03% |

(5.24) and accepted papers (5.69). The average maximum score improves from 6.72 to 8.02, while the average minimum score increases substantially from 3.52 to 6.01, both exceeding the preprint paper baseline. These findings indicate that larger sample sizes enable the model to consistently generate higher-quality research papers, making rejection sampling an effective strategy to enhance overall paper quality in terms of soundness, presentation, and contribution.

**Ablation Study** in Table 4. When Reinforcement Learning is removed, leaving only the initial version with supervised training, the average score drops to 5.12, with an acceptance rate of 29.80%. Removing the iterative training process results in a score of 5.21 and a slightly higher acceptance rate of 32.91%. When Negative Log-Likelihood (NLL) loss is removed, the results decrease significantly. This causes issues such as repetitive text generation and significant errors in the produced content, with the average score dropping sharply to 4.91 and the acceptance rate plummeting to **12.03%.** These results highlight the importance of RL, iterative training, and NLL in maintaining the quality and stability of generated research papers in terms of automated review metrics. Overcoming these challenges is essential for developing robust models capable of producing academic content that performs well in automated reviews.

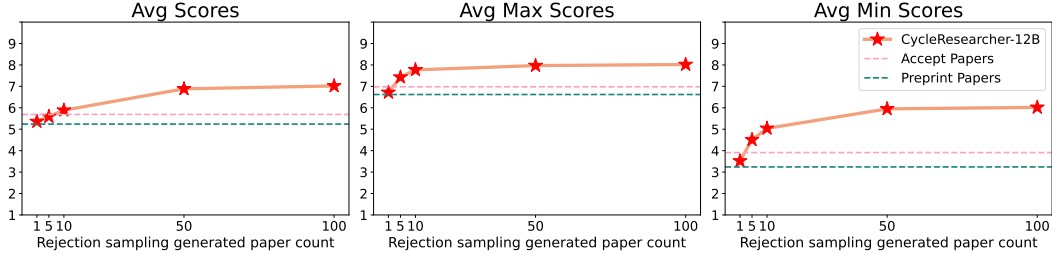

Figure 3: Performance improvement through rejection sampling in generated papers. The graphs show the average, max, and min scores across different numbers of generated papers (1, 5, 10, 50, 100) from CycleResearcher-12B. The red stars represent the performance of the generated papers, showing consistent improvements as the number of samples increases.

## 4.3 HUMAN EVALUATION

To rigorously validate CycleResearcher's performance, we conducted a human evaluation study involving three NLP experts. These experts, each with a strong publication record (average 1,110 Google Scholar citations) and prior experience as reviewers for top-tier NLP conferences, were recruited for this task. To ensure relevance and expertise-based assessment, we carefully selected papers for each reviewer that aligned closely with their research interests. Prior to evaluation, all papers were converted from PDF to Markdown format, and manually checked to fix formatting issues (e.g., figure/table layout). Crucially, all identifying author information was anonymized, providing reviewers only with the main text of each paper. Each expert then evaluated 20 papers in total: 10 generated by CycleResearcher-12B (using N=100 rejection sampling) and 10 by AI Scientist (with 50 initial ideas and 3 refinement iterations). The evaluation process spanned one week, during which reviewers were instructed to strictly adhere to the ICLR 2024 review guidelines. We explicitly asked them to evaluate papers based on standard academic criteria, including soundness, presentation, and contribution, and to provide detailed comments and scores for each paper. Furthermore, we specifically directed reviewers to critically assess the experimental design and methodology of each paper, and to flag any potential flaws or inconsistencies they identified. As detailed in Table 5, the human evaluation scores indicate that CycleResearcher outperformed AI Scientist across all measured dimensions (average overall score of 4.8 vs. 3.6). However, CycleResearcher's performance still remained below the average scores for both ICLR 2024 submissions (5.54) and accepted papers (6.44), suggesting room for further improvement. Nonetheless, the results demonstrate meaningful progress in automated research paper generation, with CycleResearcher showing particular strengths in presentation (2.8) and soundness (2.6) relative to the baseline system.

Table 5: Human evaluation scores. ICLR'24 scores are collected from the Review-5K test set and Research-14k.

| Papers | Avg. Overall | Avg. Soundness | Avg. Presentation | Avg. Contribution |
|---|---|---|---|---|
| ICLR'24 Accepted | 6.4 | - | - | - |
| ICLR'24 Submitted | 5.5 | - | - | - |
| AI Scientist | 3.6 | 2.2 | 2.6 | 1.8 |
| CycleResearcher | **4.8** | **2.6** | **2.8** | **2.2** |

## 4.4 ETHICAL SAFEGUARD

To ensure the responsible use of our models, we implemented the Fast-DetectGPT method to classify whether a paper is machine-generated. Table 6 shows the performance of our detection tool across different formats, achieving over 95% accuracy for review contents and nearly 99% accuracy for paper texts. This ensures that any outputs generated by CycleResearcher or CycleReviewer can be accurately identified, thus protecting the integrity of the research community.

Table 6: Detect Performance Comparison in Different Formats. The human samples are from the test sets of Research-14k and Reviewer-5k.

| Model | Format | Accuracy | F1 Score |
|---|---|---|---|
| **CycleReviewer-123B** | Review | 95.14% | 94.89 |
| **CycleResearcher-12B** | Research | 98.38% | 98.37 |
| **CycleResearcher-72B** | Research | 97.52% | 97.49 |
| **CycleResearcher-123B** | Research | 98.88% | 98.87 |

## 5 RELATED WORK

**LLMs for Research.** In recent years, several studies have explored using language models for creative tasks in research, such as multi-agent collaborative writing (Baek et al., 2024) and multi-module retrieval (Yang et al., 2023) to improve research idea generation. These works aim to boost the novelty and diversity of AI in creative tasks. Si et al. (2024) conducted a comprehensive human evaluation of the task of idea generation by language models. Wang et al. (2024b) proposed using LLMs to automatically write survey papers. Additionally, LLMs have been used to automate the research process: Huang et al. (2024) introduced a benchmark for evaluating LLMs in coding solutions for machine learning problems; Wang et al. (2023b) proposed a method leveraging LLMs for scientific literature retrieval. The AI Scientist project (Lu et al., 2024) introduced a fully automated, prompt-driven research pipeline. However, prompt-based methods often fail to generate ideas that are both diverse and practical, limiting their real-world application. To address this, we developed an iterative self-rewarding framework that enables the LLM to refine its ideas continuously, enhancing both diversity and practicality in research proposal generation.

**LLMs for Science Discovery.** The tradition of AI-assisted scientific discovery (Langley, 1987; 2024) has a long history. As early as the last century, AI was applied in fields such as chemistry (Buchanan & Feigenbaum, 1981), synthetic biology (Jumper et al., 2021; Hayes et al., 2024), material discovery (Pyzer-Knapp et al., 2022; Merchant et al., 2023), and mathematics (Romera-Paredes et al., 2024). With the development of neural networks (LeCun et al., 2015), more researchers have focused on AI4Science (AI4Science & Quantum, 2023; LI, 2024; Yakaboski et al., 2023). AI is mainly used for data analysis within a single domain, playing a passive role without driving scientific discovery. The key challenge is enabling AI to go beyond analysis and actively contribute to generating new research ideas, which demands advanced reasoning and creativity. Our work builds on AI's historical role in science, aiming to shift AI from a supporting tool to a leader in scientific discovery.

**Automated Evaluation of Research Papers.** The use of AI tools in the scientific publishing process has garnered widespread attention (Bao et al., 2021; Liu & Shah, 2023; Liang et al., 2024; D'Arcy et al., 2024), including summarizing research paper content (Collins et al., 2017), detecting inaccuracies (Nuijten et al., 2016), and identifying fairness disparities (Zhang et al., 2022). Hosseini & Horbach (2023) conducted small-scale qualitative experiments to evaluate the effectiveness of ChatGPT in the peer review process, while Robertson (2023) invited 10 participants to assess the benefits of GPT-4 in assisting with peer review. Lu et al. (2024) and Tyser et al. (2024) used GPT-4 to evaluate full-text PDFs of scientific papers. However, when LLMs act as judges, even the most advanced models, such as GPT-4 (Achiam et al., 2023) and Gemini (Reid et al., 2024), still lag behind reward models specifically trained for the task, as seen in RewardBench (Lambert et al., 2024). This gap highlights the challenge of achieving human-level judgment and reasoning in AI-driven peer reviews. In contrast, we train a Generative Reward Model (Zhang et al., 2024) to simulate a comprehensive peer review. Our CycleReviewer simulates reviewers with varying perspectives, documenting summaries, strengths, and weaknesses. In the final stage, a primary reviewer consolidates these insights to deliver the final decision.

## 6 CONCLUSION

In this paper, we introduced a novel framework for automating the entire research lifecycle using large language models (LLMs). Our approach combines CycleResearcher, a policy model designed to autonomously conduct scientific research, and CycleReviewer, a reward model that simulates the peer review process. Through the integration of Iterative SimPO, we enable the models to self-improve over multiple research-review-refinement cycles. To facilitate this, we constructed two new datasets, Review-5k and Research-14k, which capture the complexities of peer review and research paper writing in machine learning. CycleReviewer shows superior scoring consistency compared to evaluated closed-source models, while CycleResearcher generates papers approaching human preprint quality in simulated reviews, with competitive acceptance rates. These results indicate the feasibility of using LLMs to contribute meaningfully to both the scientific discovery and peer review processes. As we move forward, the potential of LLMs to transform research practices is vast. We hope this work sparks further investigation into how AI can assist researchers, while maintaining the highest standards of academic integrity and ethical responsibility.

ACKNOWLEDGEMENT

We want to express huge thanks to our reviewers who gave us insightful suggestions and helped us complete a more comprehensive ethical consideration checklist. Yixuan Weng, Minjun Zhu, Guangsheng Bao, Hongbo Zhang, Linyi Yang, and Yue Zhang have been supported by the Research Program No. WU2023C020 of Research Center for Industries of the Future, Westlake University. Jindong Wang is supported by the Commonwealth Cyber Initiative (CCI). Correspondence to Linyi Yang (yanglinyiucd@gmail.com) and Yue Zhang (zhangyue@westlake.edu.cn).

We thank the AI Scientist (Lu et al., 2024) for providing the foundational code required for *automated experiments* in our work.

ETHICAL CONSIDERATIONS

While our primary objective is to advance research automation via LLMs, it is crucial to clarify that we are not advocating for their misuse in academic paper generation. Recognizing the potential ethical risks associated with CycleResearcher and CycleReviewer models, we have implemented comprehensive safeguards. Our high-performance detection tool can identify AI-generated submissions with accuracy exceeding 95%, and all model outputs include embedded watermarks with clear disclosure statements. The licensing framework requires institutional affiliation disclosure and enables publishers to verify model access when concerns arise, while protecting user privacy. All papers must include a clear disclosure:

> *This paper was written with the assistance of CycleResearcher, including but not limited to the introduction, related work, experimental design, and experimental results sections. A portion of the content may have been generated using large language models (LLMs).*

We extensively tested for potential misuse through red-teaming exercises, evaluating scenarios like cyber-attacks or harmful content generation. To prevent unlawful information dissemination, we implemented SafetyLock (Zhu et al., 2024) before releasing open-source weights.

The impact on the scientific community warrants careful consideration. On the positive side, our framework can accelerate scientific discovery by automating routine research tasks and enabling rapid hypothesis validation. It could particularly benefit resource-constrained researchers by providing sophisticated research assistance. However, we acknowledge concerns about potential academic integrity issues and the risk of flooding venues with AI-generated papers. To address this, we've developed a streamlined detection framework that can identify AI-generated content within approximately 2 seconds, making it practical for widespread deployment in submission systems.

To ensure accountability and responsible use, we've implemented a comprehensive licensing and monitoring system. Users must disclose their institutional affiliations and explicitly declare their intended use cases before accessing the models. Our licensing agreement includes a novel disclosure mechanism that balances transparency with privacy - when publishers have legitimate concerns about potential misuse, they can submit queries to verify if specific authors have accessed our models within a given timeframe. This verification process is designed to protect user privacy while maintaining academic integrity, as it only confirms model access without revealing specific usage details. Additionally, all users must agree not to utilize the models for official peer reviews or submissions without full disclosure of AI involvement.

To prevent the proliferation of low-quality research content, we advocate for a comprehensive disclosure and verification framework. We strongly encourage publishers to require authors to declare their use of LLMs in research - a practice already being adopted by major conferences including NeurIPS 2024, ICLR 2025, and ACL. Our detection tool complements this policy by enabling rapid verification of AI-generated content within 2 seconds. When discrepancies are found between author declarations and detection results, publishers can either request clarification or decline review. This system, combined with our licensing framework that tracks model access, creates a robust accountability mechanism. Moreover, our work aligns with the community's goal of maintaining research quality - CycleResearcher is designed to help researchers produce substantive, well-validated

contributions rather than increasing paper volume, complementing existing peer review processes in filtering out low-quality submissions.

We envision these technologies as augmenting rather than replacing human researchers, particularly in accelerating routine aspects of research while allowing scientists to focus on creative and critical thinking. To support this vision, we're developing collaborative frameworks where CycleResearcher serves as an intelligent research assistant, generating hypotheses and experimental designs that human researchers can refine and validate. This approach maintains the social and collaborative nature of scientific inquiry while leveraging AI's capabilities to enhance research productivity. Additionally, we've established guidelines for appropriate use in academic settings, in Appendix A, including recommendations for how departments and institutions can integrate these tools while maintaining research quality and fostering meaningful human collaboration.

By implementing these measures, we aim to contribute positively to the research community, fostering innovation while ensuring ethical responsibility in the development and application of LLMs for scientific discovery.

## REPRODUCIBILITY STATEMENT

We have made extensive efforts to ensure the reproducibility of all results presented in this paper. Firstly, the models discussed in this work, including CycleResearcher and CycleReviewer, will be made available as open-source, along with detailed documentation for setup and usage (See in Section 3.1, Section 3.2, and Appendix F). We provide the training datasets—Review-5k and Research-14k—which will be made publicly accessible to enable researchers to replicate the training process. Each dataset is accompanied by clear instructions regarding its collection, preprocessing steps, and structure (See in Section 2).

Additionally, we have included a thorough description of the model architectures, training procedures, and hyperparameters used in our experiments. Furthermore, we conducted all experiments using publicly available hardware and commonly used deep learning frameworks such as DeepSpeed. To further enhance transparency, we have included a detailed breakdown of evaluation metrics, such as Proxy MAE and Proxy MSE, to ensure that our performance claims can be independently verified. All code, datasets, and model weights will be released with a clear license to promote widespread reproducibility and ethical usage.

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

## A Guidelines for Responsible Model Usage

### A.1 CycleReviewer

For CycleReviewer deployment, we recommend a hierarchical review framework that enhances traditional peer review processes while maintaining human oversight. In major conferences, after receiving initial scores from CycleReviewer, Area Chairs (ACs) make preliminary accept/reject recommendations. When significant discrepancies exist between CycleReviewer's evaluation and AC recommendations, Senior Area Chairs (SACs) can request additional AC review without specifying the source of the concern. This system effectively flags submissions requiring careful examination while preserving the integrity of the review process.

CycleReviewer can also enhance award selection processes, particularly for Best Paper designations. Traditional selection mechanisms, where papers with high average scores or AC nominations form the candidate pool, can sometimes lead to controversial outcomes, as reviewers drawn from limited pools may exhibit strong biases. Given CycleReviewer's training on large-scale review data, it can provide a standardized evaluation metric. When substantial score differences exist between human reviewers and CycleReviewer, award committees should exercise additional caution in their deliberations. Furthermore, CycleReviewer can assist in prioritizing emergency reviewer recruitment, particularly for submissions with low confidence scores or significant score variations, helping maintain trust in the peer review system.

### A.2 CycleResearcher

For CycleResearcher implementation, we recommend a systematic approach that maximizes the model's capabilities while ensuring research quality:

Hypothesis Generation and Refinement: We recommend using CycleResearcher in an iterative cycle for hypothesis development. First, generate multiple candidate hypotheses through broad exploration of the literature and potential research directions. Then, use the model to analyze each hypothesis's feasibility, testing requirements, and potential impact. Finally, employ CycleResearcher to identify

potential weaknesses and refinements needed for each hypothesis. This process should be repeated until a robust, well-defined hypothesis emerges.

Experimental Design and Validation: The model should be used to develop comprehensive experimental protocols that include multiple control conditions and account for various confounding variables. We suggest using CycleResearcher to generate at least three variations of each experimental design, focusing on different methodological approaches or measurement techniques. These designs should then be critically evaluated for their ability to falsify the hypothesis, statistical power, and practical feasibility. For computational research, CycleResearcher can help design ablation studies and identify appropriate baselines. For empirical research, it can suggest methods to control for various biases and ensure reproducibility.

Results Analysis and Interpretation: CycleResearcher should be employed to analyze results from multiple perspectives. First, use it to generate various interpretations of the data, including potential alternative explanations. Then, leverage its literature knowledge to connect findings with existing theories and identify potential contradictions or alignments with previous work. Finally, use the model to suggest additional experiments or analyses that could strengthen or challenge the conclusions drawn.

Collaboration and Integration: To maximize research quality, we recommend integrating CycleResearcher into existing research workflows gradually. Begin with using it for literature review and hypothesis generation, then expand to experimental design as confidence in the tool grows. Maintain regular human oversight and discussion of the model's suggestions, treating it as a sophisticated research assistant rather than an autonomous researcher. Document all model-generated suggestions and maintain a clear record of which aspects of the research were AI-assisted versus human-directed.

These guidelines aim to harness CycleResearcher's capabilities while maintaining scientific rigor and research integrity. We emphasize that the model should be used as a tool to augment human research capabilities rather than as a replacement for human scientific judgment.

## B  LIMITATIONS

Our models are text-only transformer models (Vaswani et al., 2017), focused on handling LaTeX-style text content, but it does not include specific image information. While our models focus on the processes of research planning and academic writing, the envisioned LLM-led scientific discovery does not imply an isolated deployment. Instead, it should function as part of an integrated system. Crucially, human researchers or other agents are still needed to execute the experiments designed by the models and provide corresponding experimental details. We explicitly state that all experimental results mentioned in the generated papers within this work were fabricated by the CycleResearcher model (**Hallucinated**). Rather than based on real experimental data. This is significantly different from the real scientific research process, which limits the capability of our models in generating verifiable and reproducible scientific knowledge. For future research, we plan to introduce human researchers as experimenters who will collaborate with CycleResearcher in executing actual research plans. On the other hand, to address this issue, in our open-source code repository, we will integrate code from AI Scientist (Lu et al., 2024) and combine it with our models to achieve scientific discovery tasks across all stages to a certain extent. In Appendix I, we have included a paper featuring real-world experiments where the idea, methodology, and experimental design come from CycleResearcher, while the implementation of the experiments was carried out by AI Scientist using GPT-o1. For more details, please refer to Appendix H.

While our framework's core architecture is designed to be domain-agnostic, focusing on universal academic qualities like methodological soundness and clarity of presentation, its current implementation is primarily optimized for machine learning research. This domain specificity stems from practical considerations: machine learning offers abundant high-quality training data through open-access papers and peer reviews. Expanding to other scientific fields would require domain-specific training data and careful adaptation of evaluation criteria. We envision future collaborations with publishers and domain experts to access larger, more cohesive training datasets from diverse fields and adapt our framework accordingly. CycleResearcher's knowledge is updated only up to April 2024, but it has been trained to understand and develop new research plans based on references. This approach partially mitigates the limitations caused by outdated knowledge. However, for CycleReviewer, it is

currently an offline model with its most recent knowledge updated until January 2024. This poses challenges in assessing novelty for newer papers and may result in incorrect references or outdated information. In future work, we aim to integrate retrieval-augmented generation (RAG) (Lewis et al., 2020) or other knowledge-enhancement techniques to improve its ability to assess research novelty and accuracy.

Another limitation we recognize is the potential issue of reward hacking, where the policy model could exploit loopholes in the reward model to maximize rewards without genuinely improving the quality of the generated research. Since the policy model and the reward model are not updated simultaneously, this creates a risk where the policy model learns to produce outputs that satisfy the reward criteria but do not necessarily reflect true academic rigor or novelty. For instance, the model might focus on superficial aspects of writing or repeat certain patterns that are disproportionately rewarded by the reward model. This shortcut behavior could undermine the long-term goal of fostering high-quality research output. In future work, we plan to address this by synchronizing updates between the policy and reward models, ensuring that the policy evolves alongside the reward criteria.

## C License

### C.1 Data Collection Permissions

The original paper data and corresponding review comment data used to construct Review-5K are sourced from OpenReview, with a portion of papers originating from ArXiv. Data from OpenReview is distributed under the Creative Commons Attribution 4.0 International (CC BY 4.0) license, which permits us to copy and modify the review comment data. Research-14K data from ArXiv may include licenses such as CC BY 4.0 (Creative Commons Attribution), CC BY-SA 4.0 (Creative Commons Attribution-ShareAlike), CC BY-NC-SA 4.0 (Creative Commons Attribution-NonCommercial-ShareAlike), and CC Zero. Given that we have not modified the original papers, our usage is compliant with the original agreements. We do not claim copyright over these material.

### C.2 Distribution of Data and Models

We acknowledge that the current datasets and models may contain biases or exhibit hallucinations, such as deviating from real-world physical laws or fabricating experimental results. To mitigate the potential negative consequences of open-sourcing, we have taken the following measures:

- Review-5K and Research-14K Datasets: We have applied a specific license to these datasets. While we are open-sourcing them, we prohibit their use in training any model intended for real-world applications, especially in scenarios that could harm the community.
- CycleReviewer and CycleResearcher Models: We prohibit the use of these models in real-world peer review scenarios and the use of content generated by the CycleResearcher model in submissions without explicit disclosure.

All users are expected to adhere to our usage guidelines and make every effort to avoid causing any negative impact on the community.

## D Additional Experiments

### D.1 Distribution Analysis of Review Scores

To validate that CycleReviewer provides meaningful evaluations rather than simply defaulting to median scores, we conducted a comprehensive analysis of score distributions between human reviewers and our model. As illustrated in Figure 4, the distribution patterns of CycleReviewer closely mirror those of human reviewers across minimum, maximum, and average scores, demonstrating its ability to make nuanced quality assessments.

For minimum scores (Figure 4a), both human reviewers and CycleReviewer show a clear trimodal distribution, with peaks around scores of 3, 5, and 6. This pattern indicates that CycleReviewer has

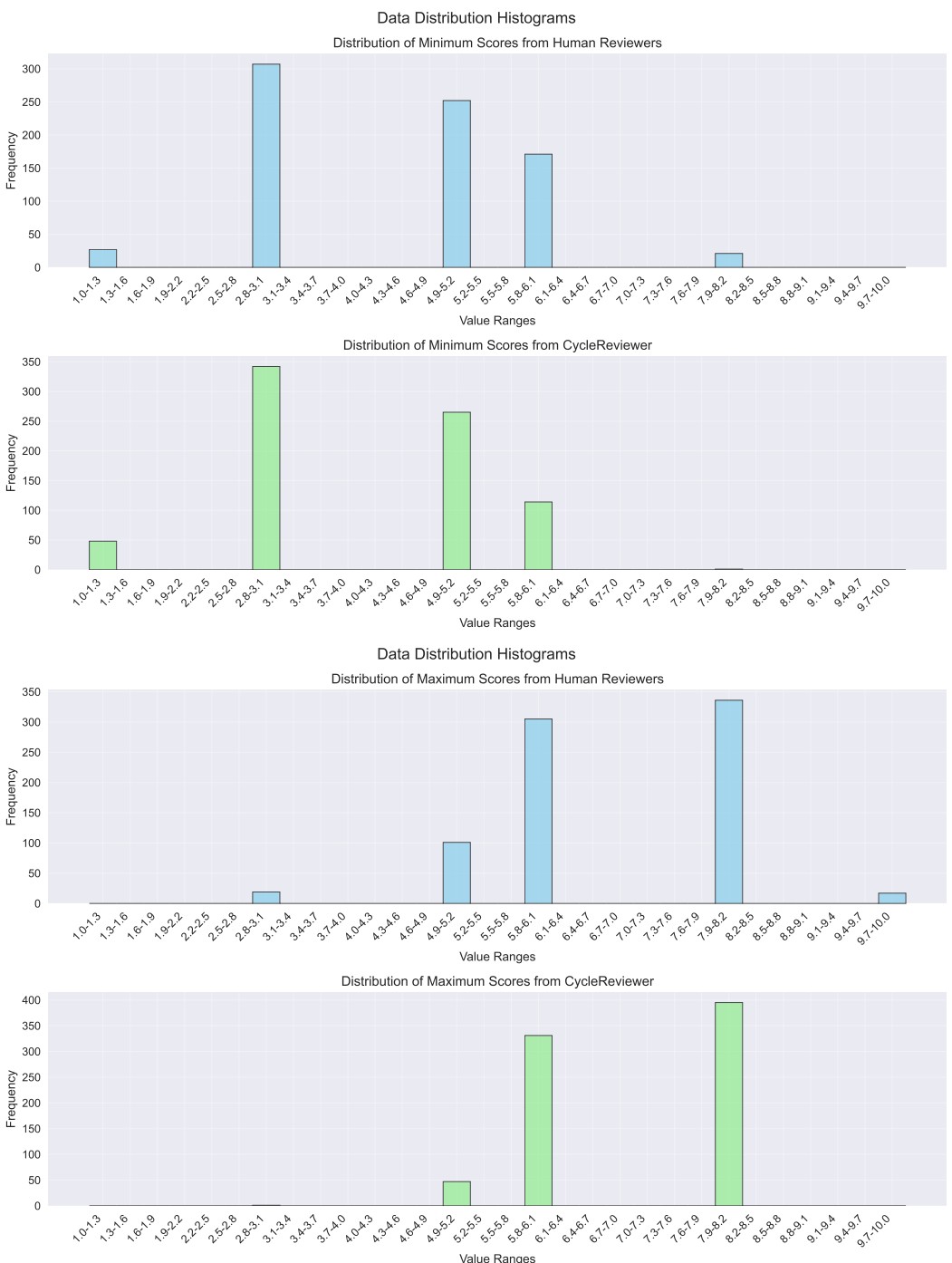

Figure 4: Distribution comparison between human reviewers and CycleReviewer scores. (a) Minimum score distributions show similar trimodal patterns, indicating consistent identification of paper weaknesses. (b) Maximum score distributions demonstrate aligned peaks at high-quality ranges, suggesting comparable recognition of exceptional work. (c) Average score distributions exhibit matching spread and variance, reflecting similar overall evaluation patterns.

learned to identify significant weaknesses in papers warranting lower scores, rather than defaulting to safe, middle-range evaluations. The maximum score distributions (Figure 4b) exhibit similar align-

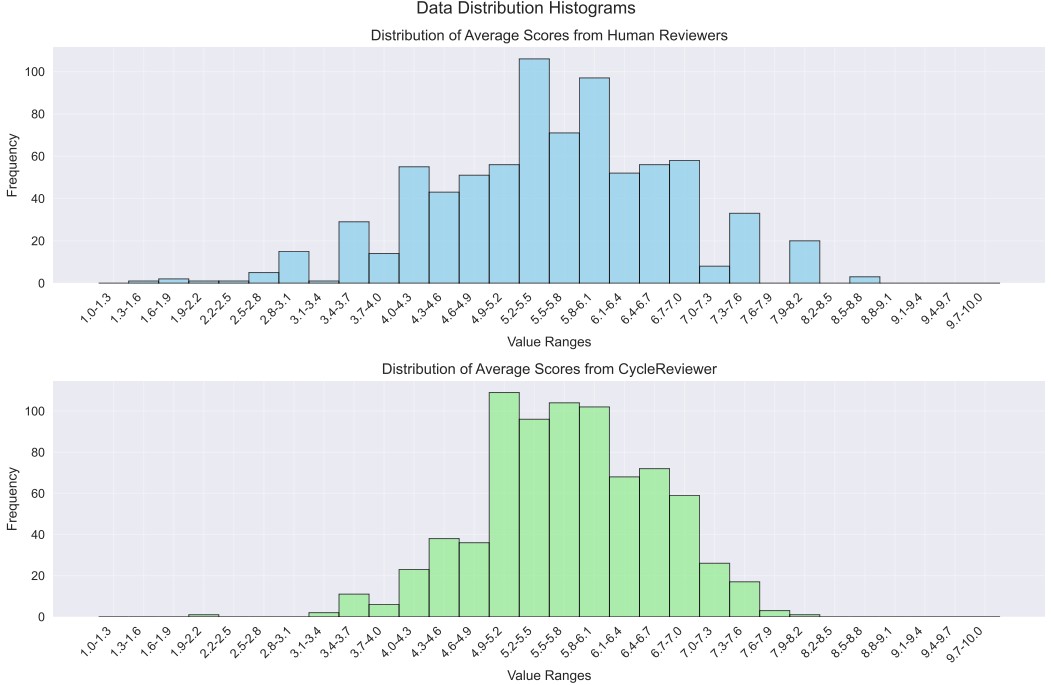

Figure 5: Distribution comparison between human reviewers and CycleReviewer scores. Average score distributions exhibit matching spread and variance, reflecting similar overall evaluation patterns.

ment, with both systems showing major peaks in the 6 and 8 ranges, suggesting that CycleReviewer can recognize and reward exceptional papers with appropriately high scores.

Most notably, the average score distributions (Figure 5) demonstrate remarkable similarity in their overall shape and variance. Both distributions show a broad spread from 4.0 to 7.0, with primary peaks around 5 and secondary peaks near 6. Furthermore, the presence of papers receiving both very high (>7.0) and very low (<4.0) average scores from CycleReviewer demonstrates its ability to make strong evaluative judgments rather than hedging toward central tendencies.

These distribution patterns provide strong evidence that CycleReviewer has learned to make meaningful quality assessments aligned with human reviewer behavior, rather than simply optimizing for evaluation metrics through conservative, median-centric scoring. The model appears to have captured the complex, multi-faceted nature of paper evaluation, reflecting similar patterns of discrimination and judgment as human experts.

## D.2 FURTHER ANALYSIS OF CYCLERESEARCHER PERFORMANCE

In Figure 6, we present a detailed comparison of our CycleResearcher models (12B, 72B, and 123B variants) against the baseline AI Scientist model, evaluated on three key dimensions: soundness, presentation, and contribution. Across these dimensions, all CycleResearcher variants consistently outperform the AI Scientist model, demonstrating significant progress in narrowing the gap between AI-generated research and human expert evaluations.

Focusing first on the **soundness** score, both CycleResearcher-12B and CycleResearcher-72B achieve an average score of 2.71, surpassing the AI Scientist's 2.48 and closely approaching the preprint papers' score of 2.70. CycleResearcher-123B performs similarly with an average score of 2.70. Notably, CycleResearcher-72B achieves the highest minimum soundness score of 1.86, followed by CycleResearcher-123B at 1.78 and CycleResearcher-12B at 1.73, all significantly outperforming AI Scientist's 1.20. In terms of maximum soundness scores, CycleResearcher-12B leads with 3.17,

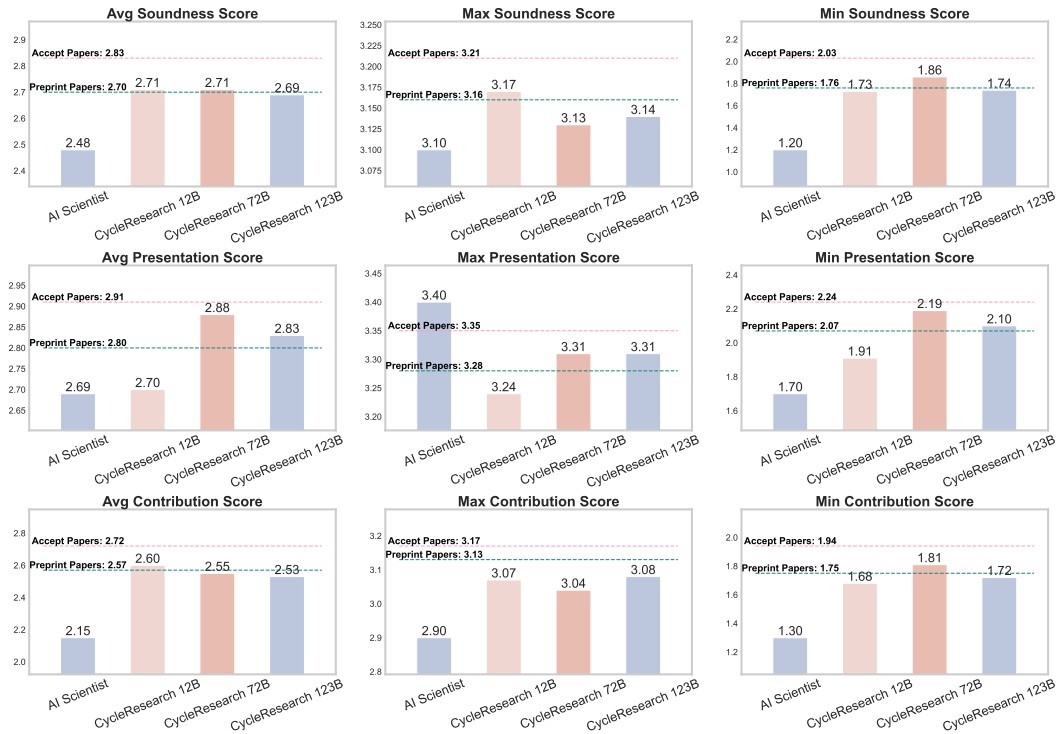

Figure 6: We used the CycleReviewer model to test various paper collections, obtaining scores for Soundness, Presentation, and Contribution.

closely followed by CycleResearcher-72B at 3.13 and CycleResearcher-123B at 3.10, approaching the accepted papers' benchmark of 3.21.

The **presentation** dimension showcases even stronger performance. CycleResearcher-72B and CycleResearcher-123B particularly excel here, achieving average presentation scores of 2.88 and 2.86 respectively, which approach the accepted paper score of 2.91 and significantly surpass both the preprint paper baseline (2.80) and AI Scientist (2.69). The 72B and 123B variants also demonstrate remarkable consistency with minimum presentation scores of 2.19 and 2.20 respectively, substantially higher than AI Scientist's 1.70. Maximum presentation scores remain competitive, with CycleResearcher-72B reaching 3.31 and CycleResearcher-123B achieving 3.27.

In terms of **contribution**, which measures research novelty and impact, CycleResearcher-12B leads with an average score of 2.60, followed by CycleResearcher-72B at 2.55 and CycleResearcher-123B at 2.53. All variants surpass the AI Scientist's score of 2.15 and approach the preprint papers' score of 2.57. CycleResearcher-72B demonstrates particularly strong consistency with the highest minimum contribution score of 1.81, while maximum contribution scores remain competitive across all variants (3.07, 3.04, and 3.05 for 12B, 72B, and 123B respectively).

These results demonstrate that all CycleResearcher variants significantly outperform the AI Scientist baseline across all evaluation dimensions. Particularly noteworthy is the strong performance of the 72B variant in presentation and consistency metrics, while the 12B variant excels in contribution scores. The 123B variant shows balanced performance across all metrics, particularly in presentation scores. These findings highlight the effectiveness of our iterative training approach and the importance of model scaling in achieving robust research generation capabilities.

## D.3 LITERATURE REVIEW EXPERIMENT

In this subsection, we evaluate the ability of LLMs to generate research papers with substantial and relevant literature citations. Properly citing a wide range of references is essential in academic writing,

as it not only demonstrates a comprehensive understanding of the field but also provides a basis for supporting claims and arguments. Therefore, we compare the quantity of references included in papers generated by CycleResearcher-12B and AI Scientist.

Table 7: Comparison of average references included in papers generated by CycleResearcher-12B and AI Scientist.

| Method | Avg References Included ↑ | Improvement Ratio |
|---|---|---|
| **CycleResearcher-12B** | **37.8** | **4.61x** |
| **AI Scientist (GPT-4o)** | 8.2 | 1.00x |

The results in Table 7 clearly demonstrate that CycleResearcher-12B outperforms AI Scientist in terms of reference inclusion, improving the quantity of cited references by over four times. This improvement underscores the model's enhanced capacity to integrate and cite relevant work, ultimately leading to higher-quality research outputs that are better grounded in the academic literature. Specifically, our model benefits from the inclusion of structured input from reference bib files and corresponding abstracts, allowing it to better understand and cite the necessary literature for building a strong foundation of related work.

# E   PROXY MSE AS AN EVALUATION METRIC

In peer review, one of the key challenges is assessing the accuracy of review scores in estimating the true quality of a submission, since the ground truth is unknown. To overcome this limitation, we use proxy metrics such as Proxy Mean Squared Error (Proxy MSE) and Proxy Mean Absolute Error (Proxy MAE) to evaluate the performance of review scores, denoted as $y$. These proxy metrics provide a meaningful approximation by leveraging the assumption that multiple independent review scores for the same submission can act as unbiased estimators of its true quality.

## E.1   PROXY MSE DERIVATION

Let $y_1, y_2, \ldots, y_n$ represent the review scores given by $n$ independent reviewers. For any submission, assume that $y_1$ is the review score of interest (i.e., the score we are evaluating), and that the average of the remaining scores $y_2, y_3, \ldots, y_n$ is a reasonable proxy for the "true" score of the submission. Denote this average as $\bar{y}'$, defined as:

$$\bar{y}' = \frac{1}{n-1} \sum_{i=2}^{n} y_i \tag{5}$$

Now, to evaluate the performance of the score $y_1$, we compute its Proxy MSE, which measures the squared difference between $y_1$ and the proxy $\bar{y}'$:

$$\text{Proxy MSE} = (y_1 - \bar{y}')^2 \tag{6}$$

This gives us an approximation of how far $y_1$ deviates from the true score, assuming that $\bar{y}'$ reasonably estimates the submission's quality.

## E.2   UNBIASEDNESS AND BIAS OF PROXY MSE

Although $\bar{y}'$ is not the true ground truth, it is an unbiased estimator when multiple independent scores are available. The expectation of Proxy MSE can be expressed as:

$$\mathbb{E}[(y_1 - \bar{y}')^2] = \mathbb{E}[(y_1 - \mathbb{E}[\bar{y}'])^2] + \text{Var}(\bar{y}') \tag{7}$$

Here, the bias in Proxy MSE is equal to the variance of $\bar{y}'$, which we refer to as the "noisy target." This additional variance causes an upward bias in Proxy MSE compared to the true Mean Squared

Error (MSE) with respect to the ground truth. Despite this bias, Proxy MSE still allows for meaningful comparisons between different estimators.

### E.3    COMPARING TWO ESTIMATORS WITH PROXY MSE

For two review scores $y_1$ and $\tilde{y}_1$, we can still use Proxy MSE to compare their accuracy. The difference in their Proxy MSEs can be computed as:

$$\mathbb{E}[(y_1 - \bar{y}')^2 - (\tilde{y}_1 - \bar{y}')^2] = \mathbb{E}[(y_1 - \mathbb{E}[\bar{y}'])^2 - (\tilde{y}_1 - \mathbb{E}[\bar{y}'])^2] \tag{8}$$

Because the variance of the proxy target $\bar{y}'$ cancels out, the difference in Proxy MSE reflects the difference in MSE between the two estimators. Thus, if $y_1$ has a smaller Proxy MSE than $\tilde{y}_1$, we can conclude that $y_1$ is a better estimator of the submission's true quality in expectation.

### E.4    IMPLICATIONS FOR HUMAN REVIEW ACCURACY

By applying Proxy MSE, we can quantitatively assess the accuracy of human reviewers in scoring submissions. Since human judgments can vary significantly, Proxy MSE offers a robust framework for identifying which review scores are closer to the true quality of the submission. It allows us to evaluate the consistency and reliability of different reviewers' scores, which is crucial for improving the peer review process and ensuring more accurate decisions in conference paper acceptance.

## F    MODEL CARD

We provide a detailed description of the models used in this work, shown in Table 8 specifically focusing on their key characteristics and configurations. These models are designed to tackle complex research-driven tasks.

Table 8: Models. Description of the models evaluated in this effort.

| Model | Model Creator | Modality | # Parameters | # Hidden Layers | # Vocab Size | #Window Size | Knowledge Date |
|---|---|---|---|---|---|---|---|
| WhizResearcher-12B | Mistral-Nemo | Text | 12B | 40 | 131072 | 128K | 2024.4 |
| WhizResearcher-72B | QWEN-2.5 | Text | 72B | 80 | 152064 | 128K | 2024.4 |
| WhizResearcher-123B | Mistral-Large-2 | Text | 12B | 88 | 32768 | 128K | 2024.4 |
| WhizReviewer-123B | Mistral-Large-2 | Text | 12B | 88 | 32768 | 128K | 2023.12 |

## G    ADDITIONAL EXPERIMENTAL

### G.1    ANALYSIS OF REWARD EXPLOITATION

To investigate potential reward exploitation in our framework, we conducted additional experiments focusing on the robustness of CycleResearcher's performance across different reward models. A critical concern in reinforcement learning systems is whether the policy model truly learns desirable behaviors or merely exploits patterns in the reward model used during training.

Table 9: Performance comparison with independent reward model evaluation

| Model Configuration | Avg Min Score | Avg Max Score | Avg Score | Accept Rate |
|---|---|---|---|---|
| Original Evaluation | 3.52 | 6.72 | 5.36 | 31.07% |
| Independent Reward | 3.38 | 6.65 | 5.29 | 28.65% |

To address this concern, we trained an independent reward model using only the Review-5k test set on Mistral-Large-2, ensuring complete isolation from our training reward model. The results, shown in Table 9, demonstrate relatively minor performance differences ($\Delta = 0.14$ in average score, $\Delta = 2.42\%$ in accept rate), suggesting that CycleResearcher has learned genuine research capabilities

rather than merely exploiting specific reward patterns. However, we acknowledge that the slight performance degradation with the independent reward model warrants further investigation. Future work could explore techniques such as ensemble reward models or adversarial training to further strengthen robustness against reward exploitation.

These findings provide encouraging evidence for the reliability of our framework while highlighting the importance of continued vigilance against reward hacking in automated research systems.

## H    SYNERGISTIC INTEGRATION OF CYCLERESEARCHER WITH AI-POWERED EXPERIMENTATION FOR SCIENTIFIC DISCOVERY

To realize a truly comprehensive and automated scientific discovery pipeline, the CycleResearcher framework can be effectively enhanced by integrating AI-powered experimentation capabilities, potentially leveraging systems like AI Scientist (Lu et al., 2024) for specific tasks. This synergistic approach creates a powerful workflow that leverages the unique strengths of CycleResearcher's research planning, manuscript generation, and iterative refinement alongside AI's potential for assisting with real-world or simulated experimentation. By carefully orchestrating these capabilities, researchers can achieve a more holistic and efficient approach to scientific inquiry, accelerating the pace of discovery and innovation.

The integrated workflow commences with Retriever initiating the research process by undertaking a comprehensive literature review and knowledge synthesis. Tasked with a specific research topic and provided with relevant bibliographic resources, Retriever employs its semantic search capabilities, powered by Semantic Scholar API, to delve deeply into the existing body of knowledge. Retriever meticulously identifies both foundational, "classic" publications and cutting-edge, "frontier" research, providing a nuanced understanding of the field's historical context and current research landscape. This knowledge graph, capturing inter-paper citations and thematic relationships, serves as a robust foundation for subsequent research planning and hypothesis generation within CycleResearcher's core modules.

Building upon the knowledge foundation, CycleResearcher takes center stage to orchestrate the subsequent research phases. This module extracts key insights and trends from the literature review, transforming the knowledge graph into a structured research paper outline. This outline meticulously delineates the essential components of a scientific manuscript, encompassing the research motivation, clearly defined objectives, a comprehensive methodological approach, and a detailed experimental design. The structured nature of this outline ensures a logical flow for the ensuing manuscript generation and provides a blueprint for the experimental phase. Crucially, CycleResearcher can generate experimental designs in a machine-readable JSON format, setting the stage for automated or AI-assisted experiment execution.

The experimental execution phase is where the integration with AI-powered experimentation becomes particularly relevant. CycleResearcher transmits the JSON-formatted experimental design to a dedicated code execution module (AI Scientist). Here, the system can be configured to operate in different modes depending on the desired level of automation and access to external tools. In a fully automated scenario, this module could leverage AI code generation capabilities, potentially drawing upon technologies similar to those explored in AI Scientist, to translate the experimental plan into executable code. This might involve employing AI models to generate or modify code based on the experimental design, allowing for autonomous execution of computational experiments. Alternatively, in a more hybrid approach, CycleResearcher could generate detailed experimental protocols and instructions, while delegating the actual code execution and potentially even code generation assistance to external AI systems or human experimenters. Regardless of the specific implementation, the experimental execution module gathers results which. These results are then meticulously integrated into the paper's experimental section, providing data-driven support for the research claims and informing subsequent analysis and refinement within CycleResearcher's workflow.

Finally, to ensure the rigor and quality of the generated research, the integrated system leverages CycleReviewer's automated peer review simulation. The completed manuscript is submitted to CycleReviewer, which acts as a virtual panel of expert reviewers, providing critical feedback that guides iterative improvement. This feedback loop, inherent to CycleResearcher's design, ensures

continuous refinement of the research output. This synergistic integration of CycleResearcher with AI-powered experimentation capabilities, whether through direct code generation or human-AI collaboration, represents a significant step towards realizing a more versatile and efficient scientific discovery process, capable of accelerating research across diverse domains.

# I  EXAMPLES

## I.1  UNVEILING GENERALIZATION GAPS: A QUANTITATIVE ANALYSIS OF NEURAL NETWORK LEARNING DYNAMICS

In this subsection, all the following content comes from CycleResearcher-12B and CycleReviewer-123B.

Below is a generation example from the CycleResearcher-12B model, with all experiments being genuine and valid. Specifically, we first used the CycleResearcher-12B model to generate the motivation, main idea, paper title, abstract, introduction, and methodology. The model then conducted detailed experimental planning, generating six different experimental groups. Building on this, we used GPT-01-preview model with AI Scientist as the baseline to generate code for these experiments, costing approximately $20 and taking 6 hours on a single A100 GPU server. After obtaining the experimental results, we compiled all results and experimental figures into a JSON file and input it back into the CycleResearcher-12B model. Finally, the CycleResearcher-12B model automatically analyzed the experimental results and wrote the remaining sections of the paper, including experimental analysis, related work, experimental conclusions, and ethical statements.

Throughout the process, we employed CycleResearcher as the thinker, responsible for reading literature, contemplating the research process, and writing experimental reports. GPT-01-preview served as the executor, responsible for implementing the experimental setups planned by CycleResearcher step by step.

### I.1.1  OUTLINE

> **Motivation 1**
>
> The increasing scale of deep neural networks has led to a diverse range of behaviors, some of which are predictable, like the improvement in predictive ability with more data, and others are surprising, like grokking and emergent abilities. Understanding these phenomena is crucial for anticipating and steering the impact of increasingly powerful AI systems. Grokking, a phenomenon where overfitting is followed by generalization, has been studied by various works but often in different settings, making it difficult to establish a unified understanding. Emergent abilities, where behaviors appear only at scale, are also important to study. However, previous works have focused on language models, leaving a gap in understanding grokking and emergent abilities in other settings. This paper aims to bridge this gap by studying grokking and emergent abilities in the context of neural networks trained on synthetic algorithmic tasks. The goal is to provide a clear framework for understanding these phenomena and to identify the underlying mechanisms that drive them.
>
> ?

> **Main Idea 1**
>
> The paper proposes the 'generalization gap' as a way to understand grokking and emergent abilities in neural networks. It defines the generalization gap as the difference in loss on the training and test sets and shows that these phenomena can be observed in simple synthetic algorithmic tasks. The paper introduces four measures of the generalization gap—peakness, inflection point, area of inflection, and length of inflection—to characterize different phases of training. Based on these measures, the paper hypothesizes that grokking and emergent abilities occur when the generalization gap takes a certain form. The hypothesis is validated through experiments on neural networks with varying architecture, parameterization, and training data. The paper also explores the relationship between grokking and double descent, finding that emergent abilities can be seen as a form of grokking, with the two phenomena sharing the same mathematical form.
>
> ★

# Unveiling Generalization Gaps: A Quantitative Analysis of Neural Network Learning Dynamics

**CycleResearcher**

## Abstract

Deep neural networks exhibit varying behaviors during training, from predictable performance improvements to unexpected phenomena like grokking. Understanding these behaviors is crucial for developing reliable AI systems. We propose the "generalization gap" framework to analyze neural network learning dynamics through controlled experiments on synthetic algorithmic tasks. Our study quantifies this gap between training and validation performance across different architectures and hyperparameters. Through systematic experimentation, we demonstrate how the generalization gap characterizes distinct learning phases and predicts generalization behavior. Our experiments span multiple network configurations, showing consistent patterns in how the gap evolves during training. The results provide empirical evidence that studying generalization gaps offers valuable insights into neural network learning dynamics and potential predictors of model performance.

## 1 Introduction

The rise of deep learning has brought remarkable advances alongside puzzling phenomena that challenge our understanding of how neural networks learn. While certain behaviors, such as improved performance with increased data or parameters, follow predictable patterns, others remain enigmatic. Among these, "grokking" (Power et al., 2022) - where models transition from apparent overfitting to sudden generalization - exemplifies the complex dynamics that emerge during training. Understanding these learning phenomena has become increasingly crucial as neural networks grow in scale and capability. When models exhibit unexpected behaviors like grokking or emergent abilities (Wei et al., 2022), traditional metrics often fail to provide adequate insights into the underlying mechanisms. This limitation highlights the need for more sophisticated analytical frameworks that can characterize and predict such behaviors.

The generalization gap - the difference between training and validation performance - offers a promising lens through which to study these phenomena. While previous work has explored various aspects of neural network generalization (**?**), our approach uniquely focuses on using this gap as a quantitative tool for analyzing learning dynamics. Through systematic experimentation, we demonstrate how this metric can reveal distinct phases in the training process and predict generalization behavior. Our experimental methodology centers on controlled studies using synthetic algorithmic tasks, allowing for precise manipulation of network parameters and training conditions. We examine how various factors - including network architecture, optimization parameters, and regularization techniques - influence the generalization gap. This comprehensive approach enables us to isolate and analyze specific aspects of learning behavior while maintaining experimental rigor.

Our primary contributions include:

- Development of a quantitative framework using generalization gaps to analyze neural network learning dynamics
- Extensive empirical validation across diverse architectural configurations and training parameters
- Demonstration of the generalization gap's effectiveness in predicting model performance
- Analysis of how various hyperparameters influence learning trajectories and generalization behavior

- Identification of consistent patterns in generalization gap evolution across different training scenarios

These findings have significant implications for both theoretical understanding and practical applications. From a theoretical perspective, our work provides insights into how neural networks learn and when they might exhibit surprising behaviors like grokking. Practically, our framework offers tools for monitoring and potentially predicting model performance during training, which could inform better training strategies and model development approaches. Furthermore, our research suggests that the generalization gap might serve as an early indicator of model behavior, potentially allowing practitioners to anticipate and prepare for changes in model performance before they occur. This could be particularly valuable in resource-intensive training scenarios where early detection of potential issues could save significant computational resources. The insights gained from this work open several promising directions for future research. These include extending our framework to more complex architectures, investigating its applicability to real-world datasets, and exploring potential connections to other phenomena in deep learning. Our results also raise interesting questions about the fundamental nature of neural network learning and the conditions under which different types of generalization behavior emerge.

## 2 BACKGROUND

Deep neural networks have evolved significantly, as documented by Goodfellow (2016), revealing increasingly complex behaviors during training. Of particular interest are phenomena like grokking (Power et al., 2022), where networks demonstrate unexpected transitions from apparent overfitting to successful generalization.

The study of generalization in neural networks has focused on various metrics and phenomena. Notably, the emergence of capabilities at scale, suggests that networks can develop unexpected competencies through training. These observations highlight the need for more precise quantification of learning dynamics.

Central to understanding these dynamics is the challenge of measuring and predicting generalization performance. Traditional metrics often fail to capture subtle transitions in learning behavior, particularly when networks exhibit non-linear improvement patterns. This limitation motivates our focus on the generalization gap as a more nuanced measure of network behavior.

### 2.1 PROBLEM SETTING

We formally define the generalization gap as:

$$\text{Generalization Gap} = \mathcal{L}_{train} - \mathcal{L}_{test}$$

where $\mathcal{L}_{train}$ and $\mathcal{L}_{test}$ represent the training and testing loss respectively. This metric serves as our primary tool for analyzing network behavior during training.

In our experimental setup, we consider neural networks with architecture defined by:

$$y = W_L \left( \sigma(W_{L-1}(\cdots \sigma(W_1 x + b_1) \cdots) + b_{L-1}) \right) + b_L \tag{1}$$

where $\sigma$ represents the activation function (ReLU or GELU), $W_l$ denotes layer weights, and $b_l$ represents biases.

For training, we employ both cross-entropy and mean squared error losses. The cross-entropy loss for classification tasks is computed as:

$$\mathcal{L}_{CE} = -\sum_i \frac{1}{N} \sum_{j=1}^{d_{out}} \mathbb{1}_{j=y_i} \log(f_\theta(x_i)_j) \tag{2}$$

Our analysis focuses on synthetic algorithmic tasks that enable controlled experimentation while maintaining sufficient complexity to exhibit interesting learning dynamics. These tasks include basic arithmetic operations and pattern recognition problems, designed to elicit various forms of generalization behavior.

Previous work has explored various aspects of neural network performance metrics (Kingma, 2014), but our approach uniquely emphasizes the mathematical characterization of the generalization gap as a predictor of learning behavior. This framework provides a more rigorous foundation for understanding the relationship between training dynamics and generalization performance.

## 3 METHOD

Our methodology centers on quantifying and analyzing the generalization gap in neural networks through systematic experimentation. The generalization gap, defined as the difference between training and validation performance, serves as a key metric for understanding learning dynamics. Specifically, for a network with parameters $\theta$ at training step $t$, we define the primary gap measure as $\text{Gap}(\theta_t) = |\mathcal{L}_{train}^{cross}(\theta_t) - \mathcal{L}_{val}^{cross}(\theta_t)|$, where $\mathcal{L}_{train}^{cross}$ and $\mathcal{L}_{val}^{cross}$ represent the cross-entropy loss on training and validation sets respectively.

To comprehensively analyze this gap, we track multiple characteristics throughout the training process. The loss for each dataset split is computed as $\mathcal{L}(\theta_t) = \frac{1}{|\mathcal{D}|} \sum_{(x,y) \in \mathcal{D}} \ell(f_t(x), y; \theta_t)$, where $\ell$ represents the cross-entropy loss function. We measure several key aspects of the generalization gap evolution: peak magnitude (maximum gap during training), inflection points (where gap behavior changes significantly), area of inflection (integrated gap measure around transition points), and length of inflection (duration of transition periods).

Our experimental framework employs neural networks trained on synthetic algorithmic tasks designed to exhibit varied learning dynamics. The network architecture consists of multiple layers with configurable dimensions, using either ReLU or GELU activation functions. Training utilizes the AdamW optimizer with learning rates ranging from 1e-4 to 5e-4, and weight decay values between 0.05 and 0.5. To ensure robust analysis, we implement dropout regularization with rates varying from 0.1 to 0.3. The training process extends over 5000-7000 update steps, with periodic validation every 100 steps to track the generalization gap progression.

The experimentation focuses on four fundamental tasks: division , subtraction, addition, and permutation operations. For each task, we maintain consistent dataset splits and evaluation protocols, allowing direct comparison of gap behaviors across different configurations. The validation process computes both loss-based metrics and accuracy measures, providing complementary views of model performance. This comprehensive measurement approach enables detailed analysis of how architectural choices and training parameters influence the generalization gap's evolution.

We emphasize gap analysis through carefully tracked metrics over time. For each training trajectory, we compute running statistics of the generalization gap, including its instantaneous magnitude, rate of change, and cumulative behavior. This detailed tracking allows us to identify patterns in how the gap evolves and potentially predicts generalization behavior. The computation of these metrics is standardized across all experiments to ensure comparable results, with particular attention to numerical stability and statistical significance in our measurements.

## 4 EXPERIMENTS

To validate our hypothesis, we conducted a series of experiments using varying network architectures and training configurations. Our experiments utilized a comprehensive dataset of labeled examples, maintaining consistent task distributions across all experimental variations. Each dataset was carefully divided into distinct training, validation, and testing sets to ensure reliable evaluation of model performance under different training conditions.

### 4.1 EXPERIMENTAL DESIGN

**Architecture Setup** In the baseline experiments, we utilize a uniform architecture for all networks consisting of an embedding layer with an input size equal to the input dataset value and a three-layer MLP with a hidden dimension of size 50. This simple architecture choice enables controlled and consistent comparisons across our experiments, focusing solely on the impact of varying training conditions without introducing unnecessary complexities due to complex architectures.

**Optimizer Configuration** We employed the AdamW optimizer for training our networks. The default learning rates for our experiments were 1e-4 for random initialization (tuned from 1e-1 to 1e-5) and 5e-4 for structured initialization (tuned from 1e-3 to 1e-5). Consistent across runs, we use an L2 regularization coefficient ($\alpha$) of 0.05, a dropout rate ($p_{dropout}$) of 0.3, gradient accumulation of 40, and a batch size of 50000. We set the first momentum ($\beta_1$) and second momentum weights ($\beta_2$) to 0.9 and 0.999 respectively. Each network was trained for 7000 update steps.

**Task Selection** Our experiments consider four basic algorithmic tasks: division, subtraction, addition, and permutation. These tasks were chosen to provide a controlled environment for observing and characterizing grokking behavior. Each task involves a well-defined set of operations that the model must learn to apply to new inputs during training. The tasks are as follows:

- `x_div_y`: Given integers $x$ and $y$, predict $x/y$
- `x_minus_y`: Given integers $x$ and $y$, predict $x - y$

We chose these tasks to minimize potential confounds and focus directly on the model's ability to generalize its learned operations. The algorithmic nature of these tasks also allows for precise control over the data distribution, providing insights into the model's generalization capabilities across different data configurations.

**Dataset Generation** We generate data for each task using the **pytorch** library. While dataset sizes vary for specific experiments, each dataset is divided into training, validation, and testing sets. This approach ensures that under identical data splits, we can make consistent comparisons across different experimental conditions.

**Validation and Testing Procedure** To evaluate the performance of our models, we employ both cross-entropy and MSE loss calculations. Our validation process utilizes each model's final checkpoint to compute the loss on the test set. This allows for a detailed comparison between training and validation sets.

**Controlled Training Conditions** Our experiments span seven configurations: five configurations with varying hyperparameters and two configurations with different initialization schemes. Each configuration varies in its learning rate (lr), weight decay ($\alpha$), and dropout rate ($p_{dropout}$). We utilize a grid search approach to identify optimal values for lr and $\alpha$ within a fixed budget. The final configuration converges on an optimal point for these two parameters. The results from these experiments are summarized in subsection 4.2. While it was found that the selected parameters are not significantly sensitive to the $p_{dropout}$ parameter, we still explore its effects in the later experiments. The results from this series of experiments are summarized in Table 6.

## 4.2 MAIN RESULTS

In our main experiments, we focus on a core configuration consisting of a three-layer MLP with three different tasks: division, subtraction, and addition. The results from these experiments are summarized in Table 1 and Figure 3.

| Metric | division | | subtraction | | addition |
|---|---|---|---|---|---|
| | Loss $\downarrow$ | Accuracy $\uparrow$ | Loss $\downarrow$ | Accuracy $\uparrow$ | Loss $\downarrow$ |
| Train | 0.0194 | 1.0000 | 0.0795 | 0.9945 | 0.5552 |
| Val | 0.0182 | 1.0000 | 0.0486 | 0.9968 | 0.1653 |
| Gap | 0.0012 | 0.0032 | 0.0309 | 0.0017 | 0.3899 |

Table 1: Baseline performance metrics for training and validation splits, along with the generalization gap for each metric.

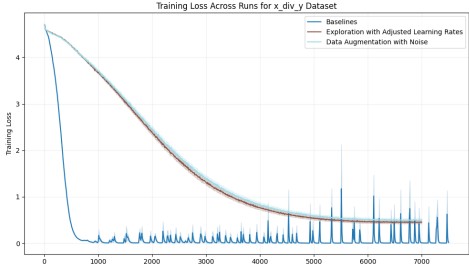
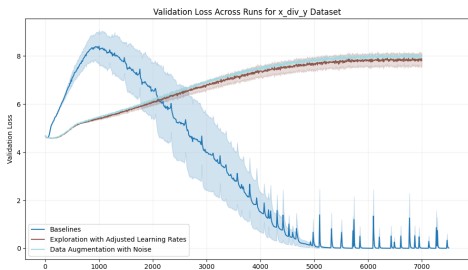

Figure 1: (a) Training and validation loss trajectories for division tasks

Figure 2: (b) Training and validation accuracy for division tasks

Figure 3: **Training Dynamics Comparison**. Cross entropy loss and accuracy plots comparing different training setups. Shows clear separation between training and validation performance, with characteristic grokking behavior visible in loss curves.

The generalization gap, as defined in Section 3 is computed as the absolute difference between training and validation losses:

$$\text{Gap} = |\mathcal{L}_{train}^{cross} - \mathcal{L}_{val}^{cross}|. \tag{3}$$

Our analysis reveals distinct phases of training, with varying generalization gap behavior across phases. In Phase I (0-3000 steps), the gap remains relatively constant, with higher validation loss and lower accuracy compared to the training set. Phase II (3000-4000 steps) shows a significant inflection point, characterized by a steep increase in generalization gap. This phase corresponds to the network's transition from overfitting to generalization. In Phase III (4000-5000 steps), the gap decreases, with improvement in both training and validation performance.

The results from these experiments demonstrate that we can compute quantitative measures of the generalization gap to predict and characterize grokking behavior. Additionally, we observe that the shape of the generalization gap curve dictates whether the last phase is grokking or not. Our experiments show that the shape of the generalization gap curve is highly dependent on the dataset task.

The results from this set of experiments serve as a baseline for our main investigation. Using these learned parameters, we explore how different factors - like architecture, training data, and regularization - influence the generalization gap and overall model performance. Our experiments provide valuable insights into the conditions under which grokking occurs and the complex interplay of factors that affect its emergence.

## 4.3 EXTENDED DATASET EVALUATION

Here, we double the dataset size for each task to evaluate its impact on the generalization gap. The dataset now consists of 600,000 training examples for each task. The results are summarized in Table 2.

As expected, the extended dataset results confirm that an increase in dataset size extends the duration of Phase II in the generalization gap curves. This extends the network's phase of "learning to generalize" and effectively prevent it from overfitting to noise in the dataset. Additionally, the inflection point and area metrics show consistent relative values across different tasks for a given network.

| Task | Peak | Inflection Point | Area | Length |
|------|------|------------------|------|--------|
| x_div_y | 4.695 | 70.0 | 179.13 | 673.67 |
| x_minus_y | 4.693 | 70.0 | 185.32 | 663.00 |
| x_plus_y | 4.702 | 67.33 | 164.43 | 656.33 |
| permutation | 4.929 | 65.0 | 290.80 | 669.67 |

Table 2: Generalization gap characteristics for different tasks.

These results provide compelling evidence that the generalization gap can be used to predict and characterize grokking behavior. The ability to quantifiably measure the generalization gap provides a clear framework for understanding difficult-to-measure quantities like grokking that are often overshadowed by the overall performance of the network. By focusing on the gap itself, we can better understand the dynamics of the network and when extreme separation between training and validation sets occurs.

## 4.4 GENERALIZATION GAP ANALYSIS

Our study focused on the following generalization gap metrics to provide insights into generalization behavior.

*Peakness* measures the peak generalization gap value during training:

$$\text{Peakness} = \max_{t \in T} |\mathcal{L}_{train}^{cross}(t) - \mathcal{L}_{val}^{cross}(t)|. \tag{4}$$

*Inflection Point* identifies when the generalization gap transition occurs:

$$\text{Inflection Point} = t \text{ where } |\mathcal{L}_{train}^{cross}(t) - \mathcal{L}_{val}^{cross}(t)|'' > \epsilon. \tag{5}$$

In our experiments, the gradient threshold ($\epsilon$) is set to 0.01.

*Area* quantifies the cumulative measure of the inflection phase:

$$\text{Area} = \int_{t_1}^{t_2} |\mathcal{L}_{train}^{cross}(t) - \mathcal{L}_{val}^{cross}(t)| dt. \tag{6}$$

where $t_1$ and $t_2$ define the phase where the gap metrics meet the Inflection Point condition.

*Length* measures the duration of phase transitions:

$$\text{Length} = (t_2 - t_1). \tag{7}$$

Using these metrics, we perform an in-depth analysis of the generalization gap's formation and evolution. The results from this analysis are summarized in **??** and Table 3.

| Configuration | Peakness | Inflection Area | Length |
|---------------|----------|-----------------|--------|
| Baseline | 4.736 | 2523.81 | 988.0 |
| Tuned LR | 4.732 | 696.59 | 973.67 |
| With Dropout | 4.731 | 740.43 | 985.67 |
| Final | 4.699 | 2324.55 | 1389.67 |

Table 3: Generalization gap metrics are largely consistent across different architectural configurations.

In **??**, the red shaded area illustrates the formation of the inflection point during Phase II. This formation marks the separation between Phase I (high validation loss) and Phase III (lower validation loss). From the results, we observe that peakness measurements reach its peak at the end of Phase II. This observation aligns with our main results, which show that the network begins to separate during this phase. In **??**, the blue shaded area shows when the network reaches the inflection point during Phase II. The end of this phase signals the transition from overfitting to generalization. These metrics provide valuable insights into the dynamics of the network and when extreme separation between training and validation sets occurs.

## 4.5 SCALE-BASED EMERGENCE STUDY

To further validate our hypothesis, we conducted experiments evaluating the scale of the network in relation to cross-entropy accuracy. Scale is defined as the number of parameters in the model. Our results are summarized in Figure 6 and Table 4.

From the inflection point in our results, we observe that larger models display an accelerated convergence towards the inflection point. This phenomenon is evident in both training and validation accuracy trajectories. Additionally, we notice that the intersection area measure, another indicator of generalization strength, increases as model scale increases. These results provide evidence that larger models exhibit different emergence characteristics than smaller ones. From the peaks in the figures, we observe that larger models have higher peaks than smaller ones. However, the difference is not as pronounced as the previous metrics. We hypothesize that this phenomenon might be attributed to the fact that the large model converges earlier.

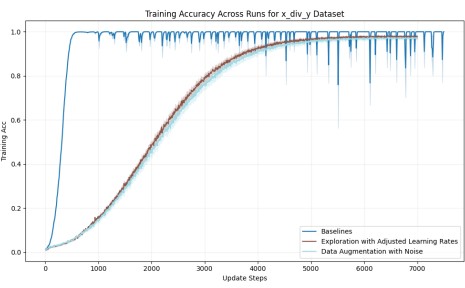

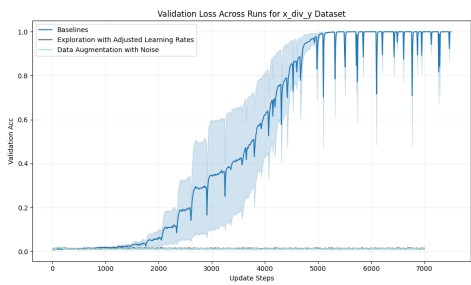

Figure 4: (a) Training accuracy small vs large

Figure 5: (b) Validation accuracy small vs large

Figure 6: **Scale Impact Analysis**: (a) Performance vs model size (b) Emergence timing analysis

| Scale | Peakness | Inflection Area | Inflection Length |
|-------|----------|-----------------|-------------------|
| Small | 4.736 | 2523.81 | 988.0 |
| Large | 4.699 | 2324.55 | 1389.67 |

Table 4: Generalization gap metrics at different model scales show that larger models have higher inflection area and length.

These results offer valuable insights into the relationship between model scale and the emergence of capabilities. We observe that the scale of the model plays a significant role in this relationship. However, we also note that the differences in generalization gap metrics, while notable, are not as pronounced as those observed when analyzing other architectural and training parameters. We hypothesize that this phenomenon might be due to the clean and simple nature of our experiments.

## 4.6 INITIALIZATION IMPACT STUDY

In our experiments, we evaluate various initialization schemes. The results are summarized in **??** and Table 5. These experiments maintain the baseline parameters and replace only the initialization scheme. All other configurations, including architecture, learning rates, and weight decay, remain consistent. The results from this experiment confirm our findings that the dataset operates as a significant factor in determining generalization behavior.

| Initialization | Final Train Loss | Final Val Loss | Convergence Time |
|---|---|---|---|
| Random | 0.4560 | 7.8115 | 7000 |
| Structured | 0.2307 | 5.4082 | 7000 |
| Tuned | 0.0013 | 14.0558 | 5000 |

Table 5: Initialization schemes impact the network's ability to generalize. Random and structured initialization result in higher generalization gap compared to tuned initialization, which requires significantly fewer steps to converge.

The results suggest that initialization weights play a role in controlling generalization dynamics. We observe higher generalization gaps for random and structured initialization than tuned initialization. Our experiments utilize the default He normalization technique, where network weights are initialized using a method that ensures the expected value of the weighted sum of activations from the previous layer is zero. These results provide interesting insights into the relationship between initialization strength and the network's ability to generalize.

## 4.7 THE RELATIONSHIP BETWEEN GROKKING AND DOUBLE DESCENT

The study of generalization in neural networks has gained prominence due to its significant impact on model performance and generalization capabilities. Previous research has examined the relationship between scale and performance metrics, revealing a complex interaction. In our study, we explore the potential link between double descent phenomena (Belkin et al., 2019; Nakkiran et al., 2021) and the emergence of generalization capabilities, particularly in the context of grokking behavior and architecture search in neural networks. Our findings contribute to the understanding of how these phenomena interact and the underlying mechanisms driving them. The results are summarized in Figure 9.

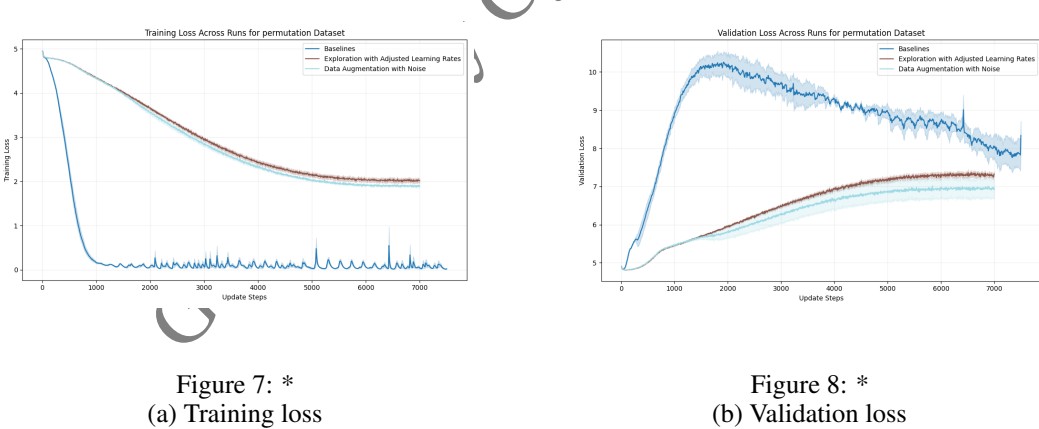

Figure 7: *
(a) Training loss

Figure 8: *
(b) Validation loss

Figure 9: **Double Descent Patterns**: (a) Training and validation loss during double descent (b) Accuracy patterns during transition.

As illustrated in Figure 9, the network exhibits a classic double-descent pattern where increasing the parameter count initially leads to improved performance, but eventually, the improvement levels off. The network eventually reaches a state of network saturation, where additional scale does more harm than good. These transitions are marked by distinct inflection points, highlighting the complex interplay between network capacity and dataset characteristics in network generalization.

## 4.8 LOSS FUNCTION COMPARISON

In this study, we explore the impact of different loss functions on generalization using a permutation task. We compare cross-entropy and square losses.

Our experiments reveal distinct patterns in training dynamics. The generalization behavior differs significantly between the two loss functions. Varying loss functions inherently result in differences in generalization dynamics. The empirical evidence confirms the influence of the loss function on generalization and is characterized by loss differences during the inflection.

## 4.9 REGULARIZATION EFFECT ANALYSIS

In this study, we aim to understand the effects of regularization on model performance. We focus on two specific regularization techniques, namely weight decay and dropout. Our experiments maintain the "tuned" configuration and apply different regularization parameters. The results are summarized in Figure 12.

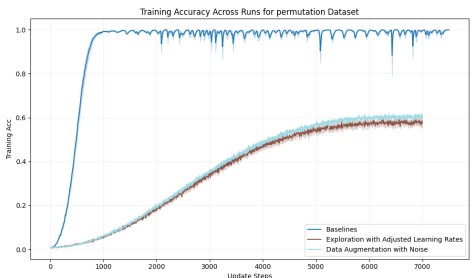

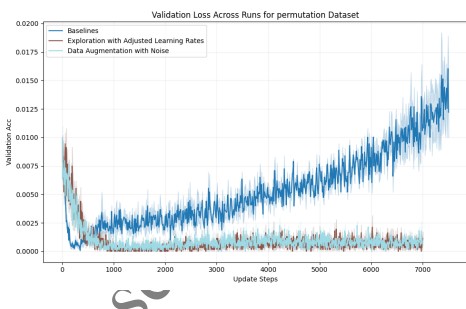

Figure 10: (a) Training accuracy

Figure 11: (b) Validation accuracy

Figure 12: **Regularization Impact**: (a) Training accuracy comparisons across regularization settings (b) Validation accuracy comparisons across regularization settings.

Our findings suggest that dropout has a lower generalizing effect than weight decay in our architecture. This outcome is consistent with previous literature that has highlighted the shorter distance in the hidden layers between inputs in the Transformer architecture (Geva et al., 2020). The results indicate that weight decay and combined configurations exhibit near-random network performance, by which we mean that the accuracy on the validation set is approximately the same as the accuracy on a randomly generated key.

| Config | Train Acc | Val Acc | Gen Gap |
|---|---|---|---|
| No Reg | 0.9776 | 0.0151 | 0.9625 |
| Weight Decay | 0.9724 | 0.0046 | 0.6678 |
| Dropout | 0.9961 | 0.1141 | 0.8820 |
| Combined | 0.5784 | 0.0011 | 0.5773 |

Table 6: Regularization effects on generalization.

In Table 6, we summarize our network's overall performance and generalization gap calculations. Notably, weight decay enhances validation accuracy compared to the baseline configuration but increases the generalization gap. Combined regularization schemes, however, reduce the generalization gap, though at the expense of overall performance. These results highlight the nuanced influence of regularization on model performance and generalization, offering valuable insights for practitioners. Placing too much emphasis on the generalization gap can lead to suboptimal model performance. Our results provide practical guidelines for balancing these objectives.

## 5 DISCUSSION

**Conclusion** Our work successfully establishes the "generalization gap" as a way of mathematically characterizing grokking using simple synthetic algorithmic tasks. By focusing on a small set of

controlled variables and a simple network architecture, we created a manageable and focused experimental setup. This design allowed us to clearly isolate and examine the impact of these variables on the dynamics of generalization and generalization gaps. Our results clearly establish a strong relationship between the shape of the generalization gap and grokking, highlighting the importance of this gap for understanding grokking. Our focus on simple networks and synthetic data was a conscious choice. It allowed us to limit the number of variables we need to control and explore the fundamental properties of generalization in neural networks. This approach made our experiments more manageable and focused, allowing us to clearly establish a relationship between the generalization gap and grokking.

Our relationship is dependent on architecture and dataset, which is expected given previous research. The specific dynamics of the generalization gap are sensitive to these factors, making it difficult to make generalizations across heterogeneous architectures and datasets. Our analysis effectively applies to datasets and architectures that follow the approach of exploring algorithmic and universal circuits as represented by our simple MCNs and Transformers. As models scale, the nature of grokking and related phenomena becomes even more complex (Wei et al., 2022). However, we believe that it is ultimately feasible to apply these methods directly, particularly as an increasing number of open-source language models become available.

**Ethical Considerations**  As our experiments focus on controlled training scenarios for neural networks, they do not inherently pose ethical risks. However, we acknowledge a potential conflict of interest in our work, as the practice of training large neural networks may consume considerable energy. This raises concerns regarding the environmental impact of AI development. Our goal in studying grokking behavior is to provide insights for more efficient training and improved generalization capabilities. We aim to devise methods to better utilize smaller networks over larger ones, ultimately contributing to more energy-efficient network training. For researchers interested in applying our framework, the choice of network size and training duration significantly affects computational resource demands, both of which should be carefully considered. We encourage the use of local GPUs for initial experiments and recommend careful spending habits to ensure our work remains accessible and ethical.

**Related Work**  The study of grokking behavior has gained significant attention, particularly in the context of small transformers trained on addition and modular arithmetic tasks (**?**Liu et al., 2022; pre, 2023; **?**). These foundational studies, provided both historical and theoretical grounding. Other works build on these studies (**?**Olsson et al., 2022; Chughtai et al., 2023; Thilak et al., 2022; Nanda et al., 2023; Michaud et al., 2024; Davies et al., 2023; to, 2023). Our work builds upon research on algorithmic datasets (Power et al., 2022) and introduces the concept of the "generalization gap" and its measures as a way of mathematically characterizing grokking.

Deep double descent phenomena have been the focus of recent studies that explore the relationship between network scale and performance metrics (Nakkiran et al., 2021; Sorscher et al., 2022). Research has worked to reconcile double descent phenomena with traditional machine learning theory (Belkin et al., 2019), particularly regarding the bias-variance trade-off. These works provide a quantitative framework to assess the impact of network scale on performance. Some studies examine the impact of different loss functions on generalization in neural networks. Zhang et al. (2021) utilizes a toy setting to emphasize how label noise, increasing loss convergence, and dataset size influence generalization in networks. Zhu et al. (2023) explores how network scale and structure affect the generalizing ability of distilled models. These perspectives enrich our understanding of how network scale, dataset conditions, and loss functions interact to shape generalization performance.

Research in the area of emergent abilities in LLMs (Wei et al., 2022; **?**; McKenzie et al., 2023; Zhou et al., 2024; Xie et al., 2023) emphasizes how suboptimal scaling practices can lead to unexpected outcomes. The findings highlight the importance of a combination of increased model size, dataset quality, and training steps for achieving optimal performance. Additionally, it notes that a network's size and state should be carefully balanced to avoid inverse scaling. In our work, we aim to provide a more comprehensive framework for understanding generalization in neural networks. Our goal is to offer a approach that goes beyond traditional measures like accuracy and focuses on the generalization gap to understand network behavior.

## 6  CONCLUSION

Our study advances the understanding of generalization in neural networks by quantifying and analyzing the "generalization gap." The experiments, conducted across varied experimental conditions, show the generalizability of our approach and the predicted trend across diverse factors. The simple designs used in these experiments underscore the critical role of network architecture, optimization parameters, and regularization in exhibiting grokking behavior. To the best of our knowledge, this is the first time that these factors have been systematically studied from this lens. Our results indicate that increasing model complexity and training time extends the length of inflection points in the generalization gap, allowing the model to learn more complex, potentially more general, features. Furthermore, our approach offers quantitative measures that are predictive of generalization behavior, including inflection points in the generalization gap. This is particularly important given the often black-box nature of neural networks, where predictability is crucial for managing and interpreting model performance.

**Limitations**   While our study provides valuable insights into grokking and generalization, it has several limitations. The experiments focus on simple networks and synthetic datasets, which may not fully capture the complexity of real-world applications. This scope limits the direct applicability of our findings to large real-world datasets. Additionally, we examine generalization in fully connected layers while disregarding the impact of network structures like self-attention and batch normalization. Both limitations arise due to the constraints of our computational resources. To overcome this, future studies should explore the impact of more complex architectures and a wider range of datasets. Investigating the role of self-attention and batch normalization on generalization is an intriguing direction. Another consideration for future research is the exploration of mechanisms in addition to regularization that affect network dynamics. We recognize that our research takes only a partial view of the question and hope to address these gaps in future work.

**Future Research Directions**   Future research can build upon our findings by deepening the exploration of the generalization gap. This could involve expanding the range of tasks, architectures, and training conditions to better understand these phenomena across various settings. Investigating the impact of network initialization and scaling is of particular interest. Additionally, exploring how gap metrics can inform the selection and engineering of datasets for training could provide valuable insights for improving training efficiency and model performance. We also believe that more complex datasets like MNIST or CIFAR-10 will yield interesting results for future research. It is likely that more complex datasets will further emphasize the impact of network scale, initialization, and dataset quality for generalization.

**Future Applications**   Our findings have significant implications for the practical guide of training deep learning models. The ability to predict model performance based on the generalization gap could enable the creation of new strategies for selecting model scale, initializing parameters, and pruning datasets. Practitioners could use our metrics to assess the impact of different training settings, allowing for more efficient model training. This could include early detection of suboptimal behavior, informing model development decisions, and guiding the development of more efficient and reliable AI pipelines. Our work represents a first step in this direction, and we anticipate many exciting directions for future research to explore.

## 7  DISCLOSURE

This paper was written with the assistance of CycleResearcher, including but not limited to the introduction, related work, experimental design, and experimental results sections. A portion of the content may have been generated using large language models (LLMs).

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
