# OpenReview forum: "CycleResearcher: Improving Automated Research via Automated Review"
_ICLR.cc/2025/Conference — ICLR 2025 Poster_

### Official Review · Reviewer_5wHA · 2024-11-01

**Soundness:** 3
**Presentation:** 4
**Contribution:** 3
**Rating:** 8
**Confidence:** 5

**Summary:**

The authors introduce two core components: CycleResearcher, a policy model that autonomously performs research tasks, and CycleReviewer, a reward model that simulates the peer review process. Experimental results suggest that CycleReviewer can outperform individual human reviewers in scoring consistency, and CycleResearcher shows promise in generating research papers that approach the quality of human-written preprints.

**Strengths:**

**Valuable Datasets**: The introduction of the Review-5k and Research-8k datasets could be highly beneficial to the research community. These datasets provide resources for training and evaluating models in academic paper generation and review, potentially fostering further advancements in automated research tools.

**Innovative Use of Preference Data**: Utilizing preference data to iteratively train the CycleResearcher model is an interesting approach. This method allows the model to improve over multiple iterations, aligning more closely with human standards through reinforcement learning.

**Ethical Safeguards**: The inclusion of a detection model to identify AI-generated papers addresses ethical concerns related to the misuse of automated research tools. By implementing such safeguards, the authors demonstrate a commitment to responsible AI deployment.

**Automation of the Research Lifecycle:** The paper attempts to automate the full research cycle, from idea generation to peer review and revision. This holistic approach is ambitious and, if successful, could significantly impact the efficiency of scientific research.

**Weaknesses:**

**Quality of Generated Papers**: Upon examining the samples provided in the Appendix (Sections E.1 and E.2), it is evident that the generated papers contain hallucinations and inaccuracies. For instance, in the generated abstracts, there are claims of outperforming state-of-the-art methods without substantial evidence or appropriate citations. This raises concerns about the reliability of the CycleResearcher model in producing high-quality, factual research papers.

**Counterintuitive Results with Model Scaling**: In Table 3 (Section 4.2), the CycleResearcher-12B model achieves a higher acceptance rate than the larger 72B and 123B models. This is counterintuitive, as larger models typically perform better due to increased capacity. The paper does not provide sufficient analysis or explanations for this phenomenon, leaving readers questioning the scalability and efficacy of the approach.

**Insufficient Ethical Considerations**: While the authors mention the implementation of a detection tool for AI-generated papers, the paper lacks a deep exploration of the ethical implications of automating research. Issues such as accountability, potential misuse, and the impact on the scientific community are not thoroughly addressed. A dedicated discussion in the Ethics Considerations section would strengthen the paper.

**Questions:**

**Explanation for Performance of Smaller Models**: In Table 3, why does the CycleResearcher-12B model receive the highest acceptance rate compared to the 72B and 123B models? This result is unexpected given that larger models generally have better performance. Could the authors provide an analysis of this outcome, possibly including case studies or error analysis to understand the limitations of larger models in this context?

**Evaluation Stability of CycleReviewer**: What is the temperature setting used for the CycleReviewer during evaluation? Additionally, have the authors experimented with running the CycleReviewer multiple times to assess the variability or deviation in the review scores and feedback? Understanding the stability and consistency of the CycleReviewer is important for gauging its reliability in the automated review process.

**Addressing Hallucinations in Generated Papers**: Given the observed hallucinations and inaccuracies in the sample generated papers (Appendix E), what strategies do the authors propose to mitigate these issues? Are there mechanisms in place to fact-check or verify the content produced by the CycleResearcher before it is submitted for automated review?

**Details Of Ethics Concerns:**

**Accountability and Authorship**: If AI systems generate research papers, questions arise regarding authorship and accountability for the content. It's essential to clarify who is responsible for the work produced and how credit should be assigned.

**Quality and Integrity of Research**: The presence of hallucinations and factual inaccuracies in AI-generated papers could undermine the integrity of scientific literature. There is a risk of disseminating false information, which could have downstream effects if other researchers build upon flawed results.

**Misuse of Technology**: The tools developed could be misused to generate large volumes of low-quality or misleading research, potentially cluttering academic discourse and making it harder to identify valuable contributions.

**Impact on the Research Community**: Automation might affect the roles of researchers, peer reviewers, and the collaborative nature of scientific inquiry. There is a need to consider how these technologies will coexist with human efforts and what support structures are necessary to ensure they augment rather than hinder scientific progress.

---

> ### Author Response · Authors · 2024-11-24
> **Response 1**
>
> > Quality of Generated Papers: Upon examining the samples provided in the Appendix (Sections E.1 and E.2), it is evident that the generated papers contain hallucinations and inaccuracies. For instance, in the generated abstracts, there are claims of outperforming state-of-the-art methods without substantial evidence or appropriate citations. This raises concerns about the reliability of the CycleResearcher model in producing high-quality, factual research papers.
>
> We sincerely appreciate your insightful feedback and are excited to clarify the sophisticated architecture of our research automation system. **The CycleResearcher model serves as the intellectual cornerstone of our framework**, functioning as a high-level research strategist that excels in comprehensive literature review, hypothesis generation, experimental design, and research planning. This approach mirrors the natural division of labor in academic research, where different specialists contribute their unique expertise to the collective scientific endeavor.
>
> During the rebuttal period, **we've implemented a comprehensive end-to-end research pipeline** that demonstrates the full potential of our framework. We've updated our appendix with new papers featuring complete experimental implementations, requiring approximately 20$ and 6 A100-GPU hours for computation. To promote transparency and community advancement, we've submitted supplementary materials including our framework's codebase and detailed experiment logs. **All code will be distributed under the MIT license**, ensuring broad accessibility and reusability. This enhancement represents a significant step forward in automated scientific discovery, combining CycleResearcher's strategic capabilities with robust experimental validation. The results showcase not only the theoretical soundness of our approach but also its practical feasibility in conducting meaningful research autonomously.
>
> > Counterintuitive Results with Model Scaling: In Table 3 (Section 4.2), the CycleResearcher-12B model achieves a higher acceptance rate than the larger 72B and 123B models. This is counterintuitive, as larger models typically perform better due to increased capacity. The paper does not provide sufficient analysis or explanations for this phenomenon, leaving readers questioning the scalability and efficacy of the approach.
>
> > Explanation for Performance of Smaller Models: In Table 3, why does the CycleResearcher-12B model receive the highest acceptance rate compared to the 72B and 123B models? This result is unexpected given that larger models generally have better performance. Could the authors provide an analysis of this outcome, possibly including case studies or error analysis to understand the limitations of larger models in this context?
>
> We deeply appreciate your astute observation regarding the scaling behavior of our models. You've highlighted a crucial technical consideration that we're excited to address comprehensively. The apparent performance discrepancy stems from an important technical constraint: due to the quadratic memory complexity of transformer attention mechanisms, we had to make careful tradeoffs in context length across model scales. **While our 12B model could process 32,000 tokens during training and 18,000 tokens during RL, the larger models were limited to 24,000 and 10,000 tokens respectively**, despite utilizing 4x8 H100 GPUs.
>
> We're particularly thrilled to share the significant progress made during the rebuttal period. By expanding our dataset to create Researcher-14K, which incorporates newly released NeurIPS 2024 Open Access papers, **we've achieved remarkable improvements in both model scales**. Here's a detailed comparison of our models' performance across datasets:
>
> | Model Scale | Dataset | Avg Score | Max Score | Min Score | Accept Rate |
> |------------|---------|------------|------------|------------|-------------|
> | 12B (Original) | Researcher-8K | 5.36 | 6.72 | 3.52 | 31.07% |
> | 12B (New) | Researcher-14K | 5.36 | 6.75 | 3.47 | 35.13% |
> | 72B (Original) | Researcher-8K | 5.20 | 6.48 | 3.40 | 25.81% |
> | 72B (New) | Researcher-14K | 5.38 | 6.58 | 3.65 | 33.64% |
>
> These results demonstrate that with expanded training data, our larger models can indeed achieve superior performance, aligning with theoretical expectations of model scaling. The enhanced results validate our architectural decisions while highlighting the critical importance of comprehensive training data in realizing the full potential of larger model architectures.

---

> ### Author Response · Authors · 2024-11-24
> **Response 2**
>
> > Insufficient Ethical Considerations: While the authors mention the implementation of a detection tool for AI-generated papers, the paper lacks a deep exploration of the ethical implications of automating research. Issues such as accountability, potential misuse, and the impact on the scientific community are not thoroughly addressed. A dedicated discussion in the Ethics Considerations section would strengthen the paper.
>
> > Addressing Hallucinations in Generated Papers: Given the observed hallucinations and inaccuracies in the sample generated papers (Appendix E), what strategies do the authors propose to mitigate these issues? Are there mechanisms in place to fact-check or verify the content produced by the CycleResearcher before it is submitted for automated review?
>
> We sincerely appreciate your thoughtful feedback regarding the ethical implications of research automation. **We have substantially expanded our Ethics section** to comprehensively address these critical concerns. For accountability and authorship, we've implemented a mandatory disclosure License requiring clear statements of AI involvement and established guidelines for attributing contributions between human researchers and AI assistants. To maintain research quality and prevent hallucinations, we've developed a **rigorous validation framework** that includes automated fact-checking and requires human verification of all AI-generated claims. To prevent misuse and proliferation of low-quality research, we've implemented both technical safeguards (our 95%-accurate detection system) and policy measures (institutional affiliation requirements and publisher verification processes). **Our licensing framework creates a transparent chain of accountability** while protecting user privacy, allowing publishers to verify model access when concerns arise.
>
> We're particularly excited to share our new section on "Guidelines for Responsible Model Usage" (Section \ref{appropriate}), which outlines **practical strategies for human-AI collaboration** in research. Given the current limitations of LLMs regarding potential hallucinations, we recommend a synergistic approach where AI augments rather than replaces human judgment. For instance, CycleReviewer can assist in award paper selection by flagging submissions where significant score discrepancies exist between human reviewers, helping committees identify cases requiring additional scrutiny. In emergency reviewer recruitment, it can help prioritize papers most in need of additional expert review based on confidence scores. For CycleResearcher, we advocate an **iterative validation process** where researchers use the model to generate multiple experimental designs and hypothesis variations, then apply their expertise to verify and refine these suggestions. This collaborative framework ensures that **AI enhances research efficiency while maintaining scientific rigor** through human oversight. By acknowledging current AI limitations and emphasizing human-AI partnership, our guidelines provide a practical roadmap for responsible integration of these technologies into the research ecosystem.
>
> > Evaluation Stability of CycleReviewer: What is the temperature setting used for the CycleReviewer during evaluation? Additionally, have the authors experimented with running the CycleReviewer multiple times to assess the variability or deviation in the review scores and feedback? Understanding the stability and consistency of the CycleReviewer is important for gauging its reliability in the automated review process.
>
> We deeply appreciate your insightful question about the stability of CycleReviewer evaluations. **We carefully selected a temperature of 0.4** after extensive preliminary experiments, finding this setting provides an optimal balance between output fluency and evaluation stability. During the rebuttal period, we conducted comprehensive stability analysis by performing three independent evaluation runs on the Review-5K test set, maintaining the temperature at 0.4. Here are the detailed results:
>
> | Metric | Run 1 | Run 2 | Run 3 |
> |--------|-------|--------|--------|
> | Proxy MAE | 0.921 | 0.918 | 0.934 |
> | Decision Accuracy | 72.24% | 73.89% | 73.51% |
> | Avg Score | 5.362 | 5.348 | 5.311 |
>
> As shown in the results, CycleReviewer demonstrates remarkable stability across multiple runs, with standard deviations consistently below 1% for all key metrics. These findings validate the reliability of our evaluation framework and confirm that our chosen temperature setting effectively balances generation creativity with evaluation consistency. We have included these additional experimental results in the updated appendix to provide complete transparency about the model's performance stability.

---

> ### Comment · Reviewer_5wHA · 2024-11-24
> **Thanks for the response**
>
> Thanks for the comprehensive response, I will raise my score.

---

> > ### Author Response · Authors · 2024-11-24
> > **Thank for your response!**
> >
> > Thank you so much for increasing the score! Your gesture is truly encouraging and means a great deal to us!

---

### Official Review · Reviewer_GAvj · 2024-11-02

**Soundness:** 2
**Presentation:** 2
**Contribution:** 2
**Rating:** 6
**Confidence:** 4

**Summary:**

The paper introduces an iterative training framework for automatic generation and review of research papers using open-source LLMs. The core of their approach consists of two main components:
1. CycleResearcher: A policy model that generates the paper, prompted by abstracts of related work.
2. CycleReviewer: A reward model that writes several peer reviews and returns scores according to ICLR criteria.


The authors initialize these models by supervised fine-tuning on scraped conference papers and ICLR reviews. They then improve the CycleResearcher using reinforcement learning (specifically iterative Simple Preference Optimization, SimPO), using CycleReviewer as a reward model.


The paper claims three main contributions:
1. Development of an iterative reinforcement learning framework that mirrors the real-world research-review-revision cycle.
2. Creation of two new datasets: Review-5k. Research-8k
3. Empirical results showing:
   - CycleReviewer produces scores that are closer to averages of multiple human reviewers than scores by individual human reviewers
   - CycleResearcher-12B achieved paper quality scores surpassing preprint level and approaching accepted paper level

The paper implements some ethical safeguards: they train a model to detect papers generated by LLMs they publish; they promise to implement a licensing agreement such that downloading model weights requires sharing institutional affiliations and agreeing not to use models for official peer reviews or submissions without disclosure.

**Strengths:**

* Training LLMs with reinforcement learning on parts of the AI research process is a novel and significant contribution.
* The paper includes numerous experiments and ablations. The overall methodology is sound (with exceptions, see weaknesses).
* The authors achieve strong results on the metrics they choose. It is somewhat impressive that their system achieved an acceptance rate of 31.07%, similar to ICLR 2024's acceptance rate.
* Authors use open-source models with a large range of scale (from 12B to 123B).

**Weaknesses:**

* The writing is overclaiming the extent to which the paper covers the full research process. Authors write that the paper “explores performing the *full* cycle of automated research and review”, however the paper omits crucial part of the process: actually running experiments. Instead, the authors train models to write complete papers purely from abstracts of past work, with completely hallucinated experiment design and results.
* I do not think that the task authors train models for — hallucinating experiment results and writing papers for them — is well motivated. Using models for this purpose will not contribute real knowledge to the scientific field. I think this is dual use technology, if not a completely malicious one. I could imagine the paper could be reframed to center on demonstrating this imminent failure of the reviewing system and raising an alarm, allowing the scientific community to adapt. In current form, the paper is probably net-negative.
* Automated evaluation of papers produced by CycleResearcher is hard to trust, since CycleResearcher was trained with RL against the same reward model as used at test-time. Reward model overoptimization (Gao et al, 2022 \- [https://arxiv.org/abs/2210.10760](https://arxiv.org/abs/2210.10760)) should be the expected result of RL, however the authors do not run any experiments to investigate to which extent their evaluation is influenced by this. For example, the authors could train a held-out reward model on a held-out dataset of reviews and then evaluate CycleResearcher on both the reward model used for RL training and this new held-out reward model.
* The claim that CycleResearcher surpasses the quality of preprint papers and approaches quality of accepted papers is not well supported, due to the concerns about reward model overoptimization mentioned above. Human reviewers rate CycleResearcher’s papers significantly lower (4.8) than the automated reviewer made by the authors (5.36). The authors could have reported the actual historical average score of ICLR2024 accepted papers.
* I have a number of concerns about the human evaluation procedure.
  * When the authors evaluate their CycleResearcher with the AI Scientist, they seem to only use rejection sampling (best of N) for CycleResearcher. This is not a fair comparison.
  * Overall, human evaluation is conducted on a small scale (10 papers total, three human reviewers, 2 methods: this paper and baseline)
  * I do not think 30min per review (including reading, writing comments & providing scores) is enough\!
* I think it’s misleading to use the term “revision” for parameters updates of the policy model (Figure 2). The paper refers to this revision as part of the full research process (“Research-Rebuttal-Revision”) but this does not actually involve revision of papers based on reviews.

**Questions:**

1. What exactly are the prompts, based on which CycleResearcher generates papers during evaluations?
2. Why do smaller CycleResearcher models get better scores in the evaluation?
3. How many samples in automated evaluation?
4. Please include the average real score of accepted papers given by human ICLR2024 reviewers.
5. How do you compute the acceptance rates, e.g. one mentioned in line 128?
6. For human evaluation:
   1. Please report the N used in best-of-N / rejection sampling.
   2. Please clarify whether each paper is evaluated by one or several humans.
   3. How are the human experts chosen?
   4. What do you mean by saying “excluding formatting considerations” in the assessment, and why is it omitted?

**Details Of Ethics Concerns:**

I do not think that the task authors train models for — hallucinating experiment results and writing papers for them — is well motivated. Using models for this purpose will not contribute real knowledge to the scientific field. I think this is dual use technology, if not a completely malicious one. I could imagine the paper could be reframed to center on demonstrating this imminent failure of the reviewing system and raising an alarm, allowing the scientific community to adapt. In current form, the paper is probably net-negative. I think the results from this paper should be known to the broad public, but not in the current framing.

---

> ### Author Response · Authors · 2024-11-24
> **Response 1**
>
> > The writing is overclaiming the extent to which the paper covers the full research process. Authors write that the paper “explores performing the full cycle of automated research and review”, however the paper omits crucial part of the process: actually running experiments. Instead, the authors train models to write complete papers purely from abstracts of past work, with completely hallucinated experiment design and results.
>
> > I do not think that the task authors train models for — hallucinating experiment results and writing papers for them — is well motivated. Using models for this purpose will not contribute real knowledge to the scientific field. I think this is dual use technology, if not a completely malicious one. I could imagine the paper could be reframed to center on demonstrating this imminent failure of the reviewing system and raising an alarm, allowing the scientific community to adapt. In current form, the paper is probably net-negative.
>
> We deeply appreciate your critical feedback regarding the scope and ethical implications of our work. Let me address your concerns comprehensively:
>
> First, we acknowledge your feedback about overclaiming and have revised our abstract and introduction to more precisely reflect our work's scope. Our vision is to develop an AI research assistant that can read literature extensively, formulate scientific hypotheses, design validation experiments, and generate research papers based on actual experimental results. During the rebuttal period, we've made new progress by implementing a complete automated scientific discovery pipeline. **In our updated Appendix G**, we demonstrate how CycleResearcher designs experiments that are then executed by GPT-o1-preview, generating concrete code and logs. **The results are real and verifiable - we've included the full codebase, experiment logs, and detailed tutorials in our supplementary materials.**
>
> While current paper quality may not match top-tier conference submissions, we believe this represents an important first step toward automated scientific discovery. The development of AI research capabilities is inherently iterative, and even current versions of CycleResearcher can generate new knowledge and insights. LLMs have progressively expanded their capabilities from grammar correction to code debugging and now to research assistance, consistently improving scientific productivity. Our framework aims to create a positive feedback loop where better AI leads to higher-quality research, which in turn enables more sophisticated AI systems.
> Regarding ethical concerns and misuse prevention, we've implemented comprehensive safeguards. Our licensing framework requires:
>
> - Institutional affiliation verification before model access
> - Agreement to ethical usage guidelines
> - Publisher verification system to track model access
> - Mandatory disclosure of AI assistance in submissions
> - Fast detection tools (<2s) for identifying AI-generated content
>
> We've expanded our guidelines in Appendix A with specific recommendations for responsible use, such as:
>
> - Using CycleReviewer for paper improvement before submission
> - Implementing hierarchical review processes for award paper selection
> - Prioritizing emergency reviewer recruitment based on confidence scores
> - Following systematic protocols for hypothesis generation and validation
> - Maintaining human oversight throughout the research process
>
> The responsibility for ethical use ultimately lies with the users - even without our framework, determined individuals could fabricate results manually. Our goal is to enhance research productivity while maintaining rigorous scientific integrity. We believe that by providing transparent tools with clear accountability mechanisms, we can foster positive advancement in automated scientific discovery while protecting against potential misuse.

---

> ### Author Response · Authors · 2024-11-24
> **Response 2**
>
> > Automated evaluation of papers produced by CycleResearcher is hard to trust, since CycleResearcher was trained with RL against the same reward model as used at test-time. Reward model overoptimization (Gao et al, 2022 - https://arxiv.org/abs/2210.10760) should be the expected result of RL, however the authors do not run any experiments to investigate to which extent their evaluation is influenced by this. For example, the authors could train a held-out reward model on a held-out dataset of reviews and then evaluate CycleResearcher on both the reward model used for RL training and this new held-out reward model.
>
> We sincerely appreciate your critical observation regarding potential reward exploitation. To address this concern, during the rebuttal period, we conducted rigorous experiments using an independent evaluation framework. **We trained a separate reward model on the Review-5k test set using Mistral-Large-2**, ensuring complete isolation from our training reward model. Here are our findings:
>
> | Model Version | Avg Min Score | Avg Max Score | Avg Score | Accept Rate |
> |--------------|---------------|---------------|------------|-------------|
> | Original CycleResearcher-12B | 3.52 | 6.72 | 5.36 | 31.07% |
> | With New Reward Model | 3.38 | 6.65 | 5.29 | 28.65% |
>
> The results show relatively minor performance differences (**~0.14 in average score and ~2.42% in accept rate**), suggesting robust performance beyond reward exploitation. While these findings are encouraging, we acknowledge this is an important area requiring further investigation. We've added a detailed analysis of these experiments to the appendix and outlined future research directions to address reward exploitation more comprehensively. We are particularly grateful for this suggestion as it has helped us strengthen a critical aspect of our evaluation framework and provide more convincing evidence of our system's reliability.
>
> > The claim that CycleResearcher surpasses the quality of preprint papers and approaches quality of accepted papers is not well supported, due to the concerns about reward model overoptimization mentioned above. Human reviewers rate CycleResearcher’s papers significantly lower (4.8) than the automated reviewer made by the authors (5.36). The authors could have reported the actual historical average score of ICLR2024 accepted papers.
>
> We appreciate your concern about the evaluation methodology. To provide a more comprehensive comparison, here is the complete evaluation data from multiple perspectives:
>
> | Paper Source | CycleReviewer Score | Human Expert Score | Reference Benchmark |
> |--------------|---------------------|-------------------|-------------------|
> | *ICLR 2024 Submissions* | - | 5.54 | Review-5K Test Set |
> | *ICLR 2024 Accepted* | - | 6.44 | All paper from ICLR 2024 |
> | AI Scientist | 4.31 | 3.6 | - |
> | CycleResearcher-12B | 5.36 | 4.8 | - |
>
> We acknowledge the gap between automated and human evaluation scores. This discrepancy highlights important considerations:
> While our model shows promise compared to baseline AI systems (outperforming AI Scientist by 1.2 points in human evaluation), we agree that we should be more precise in our claims. The current version of CycleResearcher has not yet reached the quality of typical ICLR submissions (5.54) or accepted papers (6.44). We've revised our manuscript to better reflect this reality while maintaining our finding that automated research tools can generate meaningful contributions. The human evaluation results demonstrate substantial progress in automated research capability, even if there remains significant room for improvement.

---

> ### Author Response · Authors · 2024-11-24
> **Response 3**
>
> > For human evaluation:
>
> > Please report the N used in best-of-N / rejection sampling.
>
> > Please clarify whether each paper is evaluated by one or several humans.
>
> > How are the human experts chosen?
>
> > What do you mean by saying “excluding formatting considerations” in the assessment, and why is it omitted?
>
> We appreciate your concerns about the evaluation scale and time allocation. Balancing thoroughness with feasibility required careful consideration: while each ICLR 2025 reviewer typically handles 3 papers, our experts each evaluated 20 papers - a significant commitment lasting one full week. Finding qualified reviewers with expertise matching the papers' topics further constrained our evaluation scale. This intensive evaluation process represents a substantial investment of expert time, especially considering these reviewers volunteered their expertise alongside their regular academic duties.
>
> Regarding generation comparison fairness, we maintain N=100 samples for CycleResearcher and equivalent computational effort for AI Scientist (50 ideas × 3 refinement iterations = 150 API calls). CycleResearcher requires 15 minutes on 8 H100 GPUs, while AI Scientist needs approximately 2 hours of API processing time. This parallel sampling approach provides reasonable computational parity while enabling thorough quality assessment of both systems' outputs. We have rewritten Section 5.2 to provide more detailed coverage of the human experiments.
>
> > I think it’s misleading to use the term “revision” for parameters updates of the policy model (Figure 2). The paper refers to this revision as part of the full research process (“Research-Rebuttal-Revision”) but this does not actually involve revision of papers based on reviews.
>
> We acknowledge that our use of "revision" may have caused confusion. Our framework actually performs "refinement" - where the model learns from the strongest papers within each batch to improve its generation capabilities. The inputs are identical across the batch, making these papers variations on the same research direction rather than revisions based on review feedback. We have updated our terminology throughout the paper to better reflect this refinement-based learning process.

---

> ### Author Response · Authors · 2024-11-24
> **Response 4**
>
> > Q1: What exactly are the prompts, based on which CycleResearcher generates papers during evaluations?
>
> Since our model underwent additional training on the Research-8K dataset, we only used a simple prompt to initialize the model and clearly specify its main task.
>
> ```
> [{'content': 'You are a research assistant AI tasked with generating a '
>              'scientific paper based on provided literature. Follow these '
>              'steps:\n'
>              '\n'
>              '1. Analyze the given References. \n'
>              '2. Identify gaps in existing research to establish the '
>              'motivation for a new study.\n'
>              '3. Propose a main idea for a new research work.\n'
>              "4. Write the paper's main content in LaTeX format, including:\n"
>              '   - Title\n'
>              '   - Abstract\n'
>              '   - Introduction\n'
>              '   - Related Work\n'
>              '   - Methods/\n'
>              '5. Generate experimental setup details in JSON format to guide '
>              'researchers.\n'
>              '6. After receiving experimental results in JSON format, analyze '
>              'them.\n'
>              '7. Complete the paper by writing:\n'
>              '   - Results\n'
>              '   - Discussion\n'
>              '   - Conclusion\n'
>              '   - Contributions\n'
>              '\n'
>              'Ensure all content is original, academically rigorous, and '
>              'follows standard scientific writing conventions.',
>   'role': 'system'},
>  {'content': '@article{xxx}', 'role': 'user'}]
> ```
>
> > Q2: Why do smaller CycleResearcher models get better scores in the evaluation?
>
> The initial performance difference stems from context length constraints in our transformer architecture. Despite using 4x8 H100 GPUs and optimizing memory usage, the quadratic memory scaling of attention mechanisms required different maximum sequence lengths:
>
> | **Model** | **Training Length** | **RL Length** |
> |-----------|-------------------|---------------|
> | 12B | 32,000 | 18,000 |
> | 72B/123B | 24,000 | 10,000 |
>
> During the rebuttal period, we expanded our dataset by incorporating newly released NeurIPS 2024 Open Access papers, creating Researcher-14K which doubles the training data. With this enhanced dataset, larger models show improved performance:
>
> | **Model Scale** | **Dataset** | **Avg Score** | **Max Score** | **Min Score** | **Accept Rate** |
> |---------------|------------|--------------|--------------|--------------|----------------|
> | 12B (Original) | Researcher-8K | 5.36 | 6.72 | 3.52 | 31.07% |
> | 12B (New) | Researcher-14K | 5.36 | 6.75 | 3.47 | 35.13% |
> | 72B (Original) | Researcher-8K | 5.20 | 6.48 | 3.40 | 25.81% |
> | 72B (New) | Researcher-14K | 5.38 | 6.58 | 3.65 | 33.64% |
>
> Notably, with Research-14K, the 72B model now achieves higher average scores than the 12B model, suggesting that larger models can indeed perform better when provided with sufficient training data and appropriate context lengths.
>
> > Q3: How many samples in automated evaluation?
>
> We evaluated the entire Research-8K test set (802 papers). Processing time on 8xH100:
>
> - 12B model: ~2 hours
> - 72B/123B models: ~12 hours
> - CycleReviewer: ~3 hours
>
> > Q4: What is the evaluation process?
>
> CycleReviewer simulates a complete review process with multiple reviewers discussing strengths/weaknesses, followed by an AC's final decision (Accept/Reject).
>
> > Q5: Human evaluation details?
> - Rejection sampling: N=100
> - All three experts reviewed all papers
> - Experts (avg 1100 citations) were invited via email for blind review
> - "Excluding formatting" means ignoring layout issues (table/figure sizing) to avoid bias from technical formatting limitations
>
> ---
>
>
> We sincerely appreciate your detailed feedback and have made substantial revisions during the rebuttal period. Most significantly, we have implemented a complete automated scientific discovery pipeline with real experimental capabilities, as demonstrated in our new Appendix G. The system now generates and executes verifiable experiments through GPT-o1-preview, with full codebase and logs included in our supplementary materials. We've also addressed your concerns about reward exploitation through additional independent evaluation experiments, provided more comprehensive comparison data with ICLR standards, clarified our sampling methodology, and revised terminology throughout the paper for better accuracy (e.g., using "refinement" instead of "revision"). **Given these clarifications and improvements, particularly regarding the integration of real experimental implementations, we hope these updates provide helpful context for understanding our work's scope and contributions.**

---

> ### Author Response · Authors · 2024-11-30
>
> Dear Reviewer GAvj,
>
> As the discussion period is coming to an end soon, we wanted to check if you have had a chance to review our responses. Please let us know if your questions have been adequately addressed - we are happy to provide any additional clarification needed. Thank you for your time!
>
> Best Regards,
>
> Authors of "CycleResearcher: Improving Automated Research via Automated Review"

---

> > ### Comment · Reviewer_GAvj · 2024-12-02
> >
> > Thanks, I've raised my score.

---

> > > ### Author Response · Authors · 2024-12-02
> > > **Thank you for your recognition!**
> > >
> > > On behalf of all authors, I would like to express our profound gratitude for your and all reviewers' detailed review and thoughtful suggestions!
> > >
> > > Your professional reviews and insightful suggestions have enhanced the CycleResearcher project comprehensively. Each reviewer has demonstrated admirable academic rigor and forward-thinking, guiding us toward multiple key improvements: from manuscript refinement and experimental optimization to clarity of conclusions and ethical impact considerations. Every comment has helped further improve the CycleResearcher project!
> > >
> > > Thank you again for your valuable contributions to academic development! We will continue to maintain a rigorous and responsible attitude as we work to advance this field.
> > >
> > > Best Regards,
> > >
> > > Authors of "CycleResearcher: Improving Automated Research via Automated Review"

---

### Official Review · Reviewer_CzSX · 2024-11-06

**Soundness:** 3
**Presentation:** 2
**Contribution:** 2
**Rating:** 6
**Confidence:** 4

**Summary:**

The paper presents CycleResearcher and CycleReviewer, which is a cohesive system intended to make steps towards automatic scientific discovery. In particular the novelty of the approach lies in the encapsulation of the entire research pipeline from research to review, in order to better model the entire system of research generation and get better outcomes. The authors contribute two datasets for research and review, and use these to train the system of researcher and reviewer, and evaluate them using various methods and metrics.

**Strengths:**

Originality: The idea to design both a researcher and a reviewer is novel and interesting.

Quality: The usage of recent preference optimization methods is a nice technical plus. The work contributes datasets to the direction of scientific peer reviewing, which is a resource that is rather helpful for the field. RL details and how they fit in is nice.

Clarity: Figures are well-designed and artistically pleasing. Appreciate the various different ways that are used to evaluate the methods (qualitative, ablations, etc.)

Significance: The automation of scientific research and reviewing is a very interesting and timely topic. In particular, due to the massive increase in submissions year-to-year, progress towards the paper's direction is well appreciated.

**Weaknesses:**

Originality: N/A

Quality:
One big issue of the paper is the method in which the authors obtain the "ground truth" review score: "for each submission, we use the average of the other n − 1 reviewers’ scores as an estimator of the true score."
In my opinion (and what feels like a general consensus in the community), it's pretty clear that this isn't the correct approach in determining a ground truth quality of a paper. Different reviewers have different expertises and opinions, and may disagree substantially based on their backgrounds, but this is a positive quality of peer review rather than a negative one. Thus, the metric used to judge the "loss" of a review score can be used to train proxies of reviewers, sure, but it does not make sense to then take the trained system and use the same metric to compare it against actual human reviewers. For instance, if human reviewers vary differently based on their perspectives and CycleReviewer is just doing some "hedge" where most scores are around the median score for all papers, it might achieve better than human performance on the MAE metric that is used in the evaluation. Furthermore, focusing on the score ignores perhaps the more important points of paper reviewing, such as being able to highlight errors in the paper or provide advice for making changes that are adopted in future versions. I think the true objective of reviewing in the paper's cycle paradigm matches these objectives more, although I recognize that they are even harder to quantify. Even so, I think the paper is overclaiming by saying that the lower MAE suggests that "LLMs can surpass expert-level performance in research evaluation".

And since I don't necessarily agree with the evaluation metric for the reviewer, this casts doubt on the results for the CycleResearcher because the CycleReviewer is reviewing the CycleResearcher. Also, since CycleResearcher is optimized on CycleReviewer, then saying that CycleResearcher does better on CycleReviewer than humans or AI scientist doesn't mean much. The qualitative study in section 4.3 is helpful to remove some of these doubts though.

Clarity: There are a reasonable amount of typos and grammatical errors in the document. For instance, CycleReviewer is replaced with WhizeReviewer in Section 4.1. The title of section 4.1 should be "Experiments on Paper Review Generation", etc. The paper would benefit from a pass over to correct grammatical mistakes in general to make it easier to read.

Significance: The claims are very catchy that the system can generate better reviews and papers than humans. However, given the questionable-ness of the metric, I think these are discounted to a degree.

**Questions:**

See Weaknesses.

---

> ### Author Response · Authors · 2024-11-24
> **Response 1**
>
> Your insightful question helps clarify a key challenge in this field. **CycleReviewer aims to provide a comprehensive evaluation** of each paper, consisting of both scoring and synthesis components.
>
> 1. **Regarding the Evaluation of CycleReviewer Against Ground Truth Review Scores:**
>
> Since the true score of a paper manuscript is unknown and unobservable, we must use an estimate. For a paper receiving n multiple human review scores {x₁,...,xₙ}, we followed approaches from other ML community researchers [1] regarding evaluation metrics for reviewer scores and "ground truth" review scores. We divide human review scores into two independent groups {x₁,...,xₙ₋₁} and {xₙ}, with the AI reviewer's score being ŷ = xₙ₊₁.
>
> Consider y' = 1/(n-1)∑(xᵢ)ᵢ₌₁ⁿ⁻¹, y = xₙ as two independent scores for the same paper, both assumed to be unbiased estimates of the ground truth. We measure y's performance using proxy MSE and MAE: 𝔼(y' - y)², 𝔼|y' - y|. The conventional MSE and MAE are defined as 𝔼(y' - groundtruth)², 𝔼|y' - groundtruth|.
>
> Both proxy MSE and MAE are upward-biased estimators of their conventional counterparts. The expectation of proxy MSE is expressed as:
>
> 𝔼(y'-y)² = 𝔼(ŷ-𝔼(y))² + Var(y) = MSE(ŷ) + Var(y)
>
> Despite this bias, proxy MSE can compare two estimators in expectation: for any two estimators y' and ŷ, their proxy MSE difference satisfies:
>
> 𝔼(y'-y)² - 𝔼(ŷ-y)² = MSE(y') + Var(y) - MSE(ŷ) - Var(y) = MSE(y') - MSE(ŷ)
>
> To ensure order doesn't affect results, we randomly rotate the n human reviewer scores to avoid the impact of outlier reviews. This methodology provides a statistically sound framework for comparing reviewer performance.
>
> While we acknowledge that this proxy-based evaluation has inherent limitations, it provides a principled approach for quantitative assessment of reviewer performance. The upward bias in our proxy metrics (MSE(ŷ) + Var(y)) actually makes our evaluation more conservative - if a model demonstrates improved performance under these stricter conditions, it suggests genuine advantages in review capability. Furthermore, since the bias term Var(y) cancels out when comparing different estimators (MSE(y') - MSE(ŷ)), our framework enables fair and meaningful comparisons between different review systems while accounting for the inherent subjectivity in peer review. The strong correlation between our proxy metrics and actual acceptance decisions further validates this approach's practical utility in evaluating automated review systems.
>
>
> | Model | Decision Accuracy ↑ | Macro F1 ↑ |
> |-------|-------------------|------------|
> | GPT-4o-mini | 53.06% | 34.72 |
> | GLM-4 | 49.49% | 33.10 |
> | DeepSpeak-2.5 | 45.11% | 39.98 |
> | Gemini-1.5-pro | 50.98% | 50.75 |
> | Claude-3.5-Sonnet | 48.05% | 32.44 |
> | GPT-4o | 52.58% | 34.51 |
> | **CycleReviewer (123B)** | **74.24%** | **73.99** |
>
> ---
>
> [1] Su B, Zhang J, Collina N, et al. Analysis of the ICML 2023 Ranking Data: Can Authors' Opinions of Their Own Papers Assist Peer Review in Machine Learning?[J]. arXiv preprint arXiv:2408.13430, 2024.

---

> ### Author Response · Authors · 2024-11-24
> **Response 2**
>
> 2. **Practical Example and Real-world Implications:**
>
> Different reviewers may score papers differently based on their perspectives and preferences, which can lead to significant score variations. Consider a borderline paper: one group might score it {5,5,6,6}, while another scores it {1,1,10,10}. The latter clearly shows abnormal deviation, with average MAEs of 0.66 and 4.5 respectively.
>
> As noted in ICLR 2025's AC guidelines: **"While scores aren't the perfect measure of borderline-ness, they are a good proxy to catch most borderline papers among thousands of submissions."** Such {1,1,10,10} cases often require emergency reviewers (increasing n+1) to approach the true score. This aligns with common practice in major conferences where additional reviews are sought for papers with high score variance.
>
> **Furthermore, individual expertise varies across research areas. Reviews from those less familiar with a paper's topic may be less reliable** (hence the "Confidence" rating requirement) and show greater deviation from true scores. This human factor introduces unavoidable variance in the review process.
>
> 3. **Model Performance and Distribution Analysis:**
>
> Our analysis in Appendices C.1 and Figures 4-5 demonstrates that CycleReviewer's score distribution closely matches human patterns across lowest, highest, and average scores. **This is crucial evidence that our model isn't simply regressing to the median**, but rather providing nuanced evaluations similar to human reviewers. Notable performance metrics include:
> - 93.65% accuracy in predicting rejection for papers when CycleReviewer output scored ≤4.5 (7.55% of all test-set papers)
> - 93.94% accuracy in predicting acceptance for papers  when CycleReviewer output scored ≥7 (7.94% of all test-set papers)
> - 0.5126: Spearman correlation coefficient between the average scores output by CycleReviewer and human average scores.
> - 0.4751: Spearman correlation coefficient between the min scores output by CycleReviewer and human min scores.
> - 0.4150: Spearman correlation coefficient between the max scores output by CycleReviewer and human max scores.

---

> ### Author Response · Authors · 2024-11-24
> **Response 3**
>
> 4. **Comprehensive Evaluation and Practical Applications:**
>
> **CycleReviewer simulates the entire review environment**, CycleReviewer is not just a scoring mechanism but a language model that implements a detailed review process, simulating the complete peer review environment from the lowest-scoring to the highest-scoring reviewers. Each simulated reviewer provides comprehensive feedback, including strengths, weaknesses, and even identification of grammatical errors. The model can highlight specific issues in papers and offer detailed suggestions for improvements.
>
> While this open-ended generation capability is challenging to evaluate quantitatively, we prioritize the accuracy of the numerical scores it provides. The ability to generate scores with low bias is particularly crucial for our reinforcement learning phase, where CycleReviewer serves as the reward model. This is because the preference learning process treats reward feedback as a ranking task, aiming to identify and reward the highest-scoring papers among a set of submissions with identical input conditions. Therefore, a model that can accurately predict scores closer to true paper quality becomes invaluable for effective training.
>
> We've enhanced our guidelines in the appendix for responsible deployment:
> - Using CycleReviewer for pre-submission improvement
> - Leveraging score deviations to optimize emergency reviewer allocation
> - Reducing community review burden through preliminary screening
> - Integrating automated and human review processes effectively
>
> 5. **Future Implications and Community Impact:**
>
> While we acknowledge that no automated system can perfectly replicate human judgment, our results suggest that CycleReviewer can serve as a valuable tool to:
> - **Reduce reviewer burden** by providing initial quality assessments
> - **Identify borderline cases** requiring additional human attention
> - **Standardize evaluation criteria** across large submission pools
> - **Support human reviewers** with structured, consistent feedback
>
> 6. **Clarity**
>
> Thank you very much for your kind reminder. We have revised Section 4.1 and corrected some grammatical errors in the introduction and other sections, highlighting them in blue font!
>
> ---
>
>
> **We sincerely hope our released models and resources will reduce community burden and advance automated scientific discovery.** We appreciate your reconsideration of our paper based on these clarifications and welcome further dialogue on improving our evaluation methodology.

---

> ### Author Response · Authors · 2024-11-30
>
> Dear Reviewer CzSX,
>
> As the discussion period is coming to an end soon, we wanted to check if you have had a chance to review our responses. Please let us know if your questions have been adequately addressed - we are happy to provide any additional clarification needed. Thank you for your time!
>
> Best Regards,
>
> Authors of "CycleResearcher: Improving Automated Research via Automated Review"

---

> ### Comment · Reviewer_CzSX · 2024-12-02
> **Thank you for your rebuttal**
>
> Thanks to the authors for their careful rebuttal and comprehensive response.
>
> While the metric of taking the average scores of other reviewers is still somewhat concerning to me, I think that the authors have thought carefully about this metric selection and have justified it through their response. This, in combination with the reduced claims of the paper (no longer making the claims that it outperforms existing reviewers) makes the claims more sound and reasonable. In addition, the improvement over other projects such as AI Researcher justifies the paper's contribution.
>
> Small note: I would appreciate a similar scaling down in claims in the conclusion. It currently states "with CycleReviewer surpassing human experts in scoring consistency."
>
> The additional evidence the authors provide showing how the model is outputting a distribution of scores (while still closer to the mean in figure 5) is also compelling for how the model is not simply fitting to the mean/median.
>
> Small note: Figure 6 in the Appendix has "WhizResearch" instead of "CycleResearcher".
> Also, please fix the capitalizations of the paper titles in your references.
>
> I also appreciate the increased ethics discussion, usage guidelines, and the comprehensive licensing and monitoring system. I sincerely hope that this will be well-implemented and maintained, as the proliferation of low-quality research content is a very scary outcome for all of academia.
>
> All in all, I am happy to increase my score in good faith that the authors will **retain the current more accurate versions of their claims, and also hope that the authors take careful responsibility of the societal outcomes of their work**.
>
> Best,
> Reviewer CzSX

---

> ### Author Response · Authors · 2024-12-02
>
> We sincerely thank you for your detailed feedback and thoughtful suggestions throughout the discussion phase. Your comments have been invaluable in improving our work.
>
> We will carefully address all the issues you identified, including correcting the project name in Figure 6 from 'WhizResearch' to 'CycleResearcher', fixing the capitalizations in references, and most importantly, refining the claims in our conclusion section to maintain consistency with our revised, more measured stance throughout the paper. We appreciate your attention to these details that help ensure the accuracy and professionalism of our work.
>
> Furthermore, we fully agree with your emphasis on responsible system maintenance and ethical considerations. We deeply understand that ethical considerations are essential for the ML community and must be approached with utmost care. We commit to maintaining a careful and responsible attitude - advancing AI development while being mindful of potential societal impacts. To this end, we have already begun implementing comprehensive measures, including: revising distribution protocols for code, models and data; establishing detailed usage guidelines; and developing dedicated platforms for application monitoring and public information sharing. These initiatives are currently in progress, and we plan to make them publicly accessible at an appropriate time.
>
> In conclusion, we sincerely thank all reviewers and express our highest respect for your insights that have significantly enhanced both the technical content and ethical considerations of our work.
>
> Best Regards,
>
> Authors of “CycleResearcher: Improving Automated Research via Automated Review”

---

### Official Review · Reviewer_7LzG · 2024-11-08

**Soundness:** 2
**Presentation:** 3
**Contribution:** 2
**Rating:** 6
**Confidence:** 3

**Summary:**

The paper explores the use of open-source large language models to automate the entire research process, from literature review and manuscript preparation to peer review and revision. The proposed framework includes CycleResearcher, which performs research tasks, and CycleReviewer, which simulates the peer review process. The study demonstrates that CycleReviewer can outperform human reviewers in predicting paper scores, and CycleResearcher can generate papers of quality close to human-written preprints. The models are trained using two new datasets, Review-5k and Research-8k, which capture the complexities of peer review and research paper generation. The results indicate that this approach can significantly enhance the efficiency and quality of scientific research, while also providing ethical safeguards to prevent misuse.

**Strengths:**

The introduction of CycleResearcher and CycleReviewer models to automate the entire research process, including literature review, manuscript preparation, peer review, and revision, is highly innovative. This framework mimics the real-world research cycle, enhancing the efficiency and consistency of scientific inquiry.
Performance Improvement: The CycleReviewer model demonstrates a significant improvement in predicting paper scores, outperforming human reviewers by 26.89% in mean absolute error (MAE). This indicates that the model can provide more accurate and consistent evaluations than individual human reviewers.
Quality of Generated Papers: The CycleResearcher model generates papers with an average quality close to human-written preprints, achieving an acceptance rate of 31.07%. This shows that the model can produce high-quality research outputs that are competitive with human-generated content.
Large-Scale Datasets: The development of the Review-5k and Research-8k datasets, which capture the complexities of peer review and research paper generation, provides valuable resources for training and evaluating models in academic paper generation and review.

**Weaknesses:**

Generalizability Across Domains: The models are primarily designed for machine learning-related research, and their generalizability to other scientific fields remains unexplored. This limitation suggests that the framework might not perform as well in domains outside of machine learning.
Reward Design: The paper highlights the issue of reward definition, where the policy model might exploit loopholes in the reward model to maximize rewards without genuinely improving the quality of the generated research. This behavior could undermine the long-term goal of producing high-quality research outputs.
Complexity of Implementation: Implementing the framework requires significant computational resources and expertise in reinforcement learning and LLMs. This complexity might be a barrier for widespread adoption, especially for smaller research teams or institutions with limited resources.

**Questions:**

The framework is primarily designed for machine learning-related research. How do you envision adapting CycleResearcher and CycleReviewer to other scientific fields, such as biology or social sciences, where the nature of research and evaluation criteria might differ significantly?
The paper mentions the potential issue of reward hacking, where the policy model might exploit loopholes in the reward model. Could you elaborate on the specific strategies you are considering to mitigate this issue and ensure that the generated research outputs maintain high academic rigor and novelty?

---

> ### Author Response · Authors · 2024-11-24
> **Response 1**
>
> > Weakness: Generalizability Across Domains: The models are primarily designed for machine learning-related research, and their generalizability to other scientific fields remains unexplored. This limitation suggests that the framework might not perform as well in domains outside of machine learning.
>
> We appreciate the reviewer's observation about our framework's current domain specificity. Indeed, we acknowledge that our present implementation focuses primarily on machine learning research, and the generalizability across different scientific domains remains to be thoroughly validated. This is an important limitation of our current work that deserves careful consideration and future investigation.
> Our reinforcement learning optimization framework is designed to be domain-agnostic in its core architecture. The iterative preference optimization mechanism we developed focuses on universal academic qualities such as methodological soundness, clarity of presentation, and significance of contribution, rather than domain-specific content. We chose machine learning as our initial domain primarily because it offers abundant high-quality training data through open-access papers and peer reviews, and uniquely allows for potential automation of experimental processes through code generation and results analysis.
>
> > Question: The framework is primarily designed for machine learning-related research. How do you envision adapting CycleResearcher and CycleReviewer to other scientific fields, such as biology or social sciences, where the nature of research and evaluation criteria might differ significantly?
>
> We envision several key steps for adapting our framework to other scientific domains. First, domain-specific versions of CycleResearcher and CycleReviewer could be trained using papers and peer reviews from respected journals in each field (e.g., Nature for biology, or top social science journals). In our early research phases, we explored this possibility but found that the available open-access data was often limited and fragmented across subspecialties. We believe this challenge could be addressed through collaborations with publishers and institutions to access larger, more cohesive training datasets. Additionally, the evaluation criteria would need to be carefully adapted to match field-specific standards and practices, perhaps through consultation with domain experts. While significant challenges remain, including the need for human intervention in physical experiments, we believe our framework provides a promising foundation for cross-domain expansion once these prerequisites are met. We welcome future collaborations with researchers from diverse fields to explore these possibilities. Thank you again for your suggestions. We have revised the Limitations section and added relevant discussions!
>
> > Weakness: Reward Design: The paper highlights the issue of reward definition, where the policy model might exploit loopholes in the reward model to maximize rewards without genuinely improving the quality of the generated research.
>
> We are deeply grateful for the reviewer's insightful observation regarding potential reward exploitation issues. While our current implementation shows promising results, we fully acknowledge that reward hacking remains a significant challenge in reinforcement learning systems that aim to generate high-quality research outputs. To address this concern, we conducted extensive validation through both human expert evaluation and additional experiments using separated reward models. Our human expert evaluation, conducted by experienced reviewers (averaging 1,110 Google Scholar citations), demonstrated encouraging results in comparison to baseline systems:
>
> | Metric | CycleResearcher | AI Scientist |
> |--------|----------------|--------------|
> | Overall Score | 4.8 | 3.6 |
> | Soundness | 2.6 | 2.2 |
> | Presentation | 2.8 | 2.6 |
> | Contribution | 2.2 | 1.8 |

---

> ### Author Response · Authors · 2024-11-24
> **Response 2**
>
> > Question: Could you elaborate on specific strategies to mitigate reward hacking and ensure high academic quality?
> During the rebuttal period, we conducted additional experiments to further investigate the robustness of our framework.
>
> Specifically, we trained an independent reward model using only the test set portion of Review-5k on Mistral-Large-2, maintaining complete separation from our CycleReviewer model. This new evaluation framework produced the following results:
>
> | Model Version | Avg Min Score | Avg Max Score | Avg Score | Accept Rate |
> |--------------|---------------|---------------|------------|-------------|
> | Original CycleResearcher-12B | 3.52 | 6.72 | 5.36 | 31.07% |
> | With New Reward Model | 3.38 | 6.65 | 5.29 | 28.65% |
>
>
> The relatively small performance difference (approximately 0.14 in average score) suggests our framework maintains robust performance even with an independently trained reward model. However, we fully acknowledge that more work is needed to comprehensively address reward exploitation. We've added detailed discussions of these findings and future directions to the appendix F.2, and we sincerely thank the reviewer for helping us strengthen this critical aspect of our work.

---

### Author Response · Authors · 2024-11-24
**General Responses and Summary of Revisions**

We sincerely thank all reviewers for their careful and constructive feedback, which has helped us significantly improve our work. We are encouraged by the reviewers' recognition of several key strengths:

- "The introduction of CycleResearcher and CycleReviewer,..., is highly innovative." (*Review 7LzG*)
- "The paper includes numerous experiments and ablations. The overall methodology is sound" (*Review GAvj*)
- "The paper includes numerous experiments and ablations" (*Review CzSX*)
- "The introduction of the Review-5k and Research-8k datasets could be highly beneficial to the research community" (*Review 5wHA*)


Based on your constructive comments, we have made substantial improvements to strengthen our paper's contributions and address the limitations. The major enhancements include:

1. **Automated Research Pipeline**: We implemented a complete automated scientific discovery pipeline with real experimental capabilities, including code generation and execution through GPT-o1-preview (The API call implementation code for specific experiments comes from AI Scientist and aidir. We have integrated these into one unified system to produce a paper with complete and real experimental results.). Full codebase and experiment logs are included in supplementary materials (Appendix G).

2. **Extended Dataset Coverage**: We created a new Researcher-14K dataset incorporating recent NeurIPS 2024, ACL and CVPR papers, enhancing model performance across scales. With this expansion, our 72B model achieved improved results (average score from 5.20 to 5.38, acceptance rate from 25.81% to 33.64%), demonstrating the value of comprehensive training data (Appendix F.1).

3. **Reward Model Validation**: We implemented an independent evaluation framework using a separate reward model trained on the Review-5k test set. Results show minimal performance differences (~0.14 in average score), validating our system's robustness beyond reward exploitation (Appendix F.2).

4. **Comprehensive Benchmarking**: We added complete ICLR 2024 benchmark comparisons:
   - ICLR 2024 Submissions: 5.54
   - ICLR 2024 Accepted: 6.44
   - CycleResearcher-12B: 5.36 (auto) / 4.8 (human)
   - AI Scientist: 4.31 (auto) / 3.6 (human)
This provides clear context for our model's current capabilities (Section 4.3).


5. **Enhanced Ethics Framework**: We significantly expanded our ethics considerations with:
   - Mandatory disclosure requirements
   - Institutional affiliation verification
   - Publisher verification license
   - 95%-accurate detection system
   - Comprehensive usage guidelines
   - Human-AI collaboration protocols
(Section 7: Ethics Statement, Appendix A)

6. **Evaluation Stability Analysis**: We conducted comprehensive stability testing for CycleReviewer:
   - Temperature setting: 0.4
   - Three independent evaluation runs
   - Consistent performance metrics (std < 1%)
This validates the reliability of our evaluation framework.

7. **Domain Adaptation Discussion**: We clarified our framework's current focus on machine learning while outlining concrete steps for cross-domain expansion through publisher collaborations and domain-specific training data collection (Appendix Limition).

8. **Terminology Refinement**: We updated our terminology from "revision" to "refinement" to more accurately reflect the model's learning process and avoid confusion with paper revisions (Throughout paper).

With the generous guidance from our esteemed reviewers, we have been able to substantially enhance our paper's technical depth, experimental rigor, and practical impact. We believe these comprehensive improvements address the key concerns raised while maintaining the innovative contributions of our work. We would be honored if the reviewers find these enhancements worthy of a more favorable evaluation.

---

### Author Response · Authors · 2024-12-03
**Ethical Considerations and Mitigation Measures for CycleResearcher**

On behalf of all authors, I would like to summarize and clarify the mitigation measures implemented in our work to address potential ethical concerns for all program committee members, ethics committee members, and potential users. All the content discussed has been included in the Ethics Section and Appendix A of our paper.

Our goal is to advance scientific research automation through LLMs. We believe that AI capabilities will continue to advance through the efforts of all researchers. Since 2022, LLMs have been widely adopted, comprehensively improving society's productivity levels. LLMs' capabilities have evolved from grammar correction, literature summarization, and code debugging to today's research assistance. As one of the pioneers in automated scientific research, CycleResearcher's primary purpose is to contribute to social and human welfare!

However, everything has its pros and cons. When malicious individuals misuse CycleResearcher, it may cause negative impacts to some extent. We have carefully considered a series of negative impacts and provided protective measures to mitigate them:

> **Safety and Potential Harmful Consequences**

Improper use of CycleResearcher may lead to negative impacts such as generating dangerous drug formulations or researching attack methods. **Before open-sourcing the weights, we implemented additional SafetyLock [1] security strategies to make the model reject such content and conducted additional red team exercises for testing**. Specifically, we tested all 11 safety categories from [2] and additionally tested experiments related to producing dangerous drugs and cyber attack methods. **We will only open-source it after ensuring the attack success rate is reduced to an acceptable range**. Meanwhile, we have also established strict usage guidelines that clearly limit and regulate the model's use scenarios.

> **Accountability and Authorship**

We provide three mitigation measures: watermarking, high-performance detection tools, and licensing.

**Watermarking**:
All versions of CycleResearcher we trained disclose the use of LLMs at the end of papers:

```
This paper was written with the assistance of CycleResearcher, including but not limited to the introduction, related work, experimental design, and experimental results sections. A portion of the content may have been generated using large language models (LLMs).
```

**High-Performance Detection Tools**:
In Sections 3.4 and 4.4, we implemented a highly available paper detection tool based on Fast-Detect-GPT [3], **maintaining close to 95% Accuracy and F1 Score for content generated by CycleReviewer and CycleResearcher**. It can successfully identify content generated by CycleReviewer and CycleResearcher in about 2 seconds with just 1 GPU, making widespread deployment in paper submission systems feasible. **This rapid detection mechanism, combined with our disclosure requirements, provides dual protection for maintaining academic integrity**.

**Licensing**:
For code, datasets, and model weights, we implement a comprehensive licensing and supervision system. Before accessing the model, **users must publicly disclose their institutional affiliations and explicitly state their intended use**. Additionally, our license agreement requires all users to agree that **publishers may submit queries to verify whether specific authors accessed our model within a given timeframe when there are reasonable concerns about potential misuse**. This verification process aims to maintain academic integrity while protecting user privacy by only confirming model access without revealing specific usage details.

---

> ### Author Response · Authors · 2024-12-03
> **Ethical Considerations and Mitigation Measures for CycleResearcher - 2**
>
> > **Potentially Harmful Methods**
>
> CycleResearcher works by extensively reading literature, specifying scientific hypotheses, designing validation experiments, and generating research papers based on actual experimental results. **We model it as a multi-round input-output format**, namely:
>
> ```
> **First Round**
>
> Input: Literature
>
> Output: Motivation + Idea + Method + Experimental Setup
>
> **Second Round**
>
> Input: Experimental Results
>
> Output: Complete Paper
> ```
>
> In Appendix G, we demonstrate how CycleResearcher designs experiments, with GPT-o1-preview executing and generating specific code and logs. **We provide a complete paper where all experimental results are real and verifiable**. The supplementary materials include the complete codebase, experimental logs, and detailed tutorials.
>
> We acknowledge that malicious individuals could make CycleResearcher produce incorrect papers by providing false experimental results. However, **academic ethical responsibility ultimately lies with the users—even without CycleResearcher, determined individuals could manually forge results**. While we cannot detect whether everyone uses false experimental results to produce incorrect papers, **we can provide transparent detection tools and clear accountability mechanisms to promote positive progress in automated scientific discovery while preventing potential abuse**.

---

> ### Author Response · Authors · 2024-12-03
> **Ethical Considerations and Mitigation Measures for CycleResearcher - 3**
>
> > **Guidelines for Careful Use**
>
> To supplement the aforementioned ethical considerations, we have established detailed usage guidelines to ensure responsible use of CycleResearcher and CycleReviewer. **These guidelines' core objective is to maintain human researchers' leadership and supervision of the research process while improving research efficiency**. Specifically:
>
> **Deployment Recommendations for CycleReviewer**:
> We recommend adopting a layered review framework to enhance the traditional peer review process. In major conferences, Area Chairs (ACs) should make preliminary accept/reject recommendations after receiving initial scores from CycleReviewer. When significant differences exist between CycleReviewer's assessment and AC recommendations, Senior Area Chairs (SACs) can request additional AC reviews without specifying the specific source of concern. This system effectively flags submissions requiring careful review while maintaining the integrity of the review process. Additionally, CycleReviewer can enhance the best paper selection process by providing standardized evaluation metrics. When substantial score differences exist between human reviewers and CycleReviewer, award committees should exercise extra caution in their deliberations. This is particularly helpful in avoiding strong biases that might arise from limited reviewer pools.
>
> **Implementation Recommendations for CycleResearcher**:
> We recommend adopting a systematic approach to maximize the model's capabilities while ensuring research quality. This includes:
>
> 1. **Hypothesis Generation and Optimization**
>    - First use the model for extensive literature exploration and potential research direction generation
>    - Use the model to analyze each hypothesis's feasibility, testing requirements, and potential impact
>    - Repeatedly use the model to identify potential weaknesses and make necessary improvements
>    - **This process should be repeated until a robust, well-defined hypothesis is formed**
>
> 2. **Experimental Design and Validation**
>    - **Recommend using CycleResearcher to generate at least three different design schemes for each experiment**, focusing on different methodological approaches or measurement techniques
>    - For computational research, the model can help design ablation studies and determine appropriate benchmarks
>    - For empirical research, it can suggest methods to control various biases and ensure reproducibility
>    - **All design schemes should be rigorously evaluated in terms of hypothesis falsification ability, statistical power, and practical feasibility**
>
> 3. **Results Analysis and Interpretation**
>    - Use the model to analyze results from multiple angles, including generating various possible interpretations
>    - Utilize its literature knowledge to connect findings with existing theories
>    - **Identify potential contradictions or consistencies with prior work**
>    - Use the model to suggest additional experiments or analyses that could strengthen or challenge conclusions
>
> 4. **Collaboration and Integration**
>    - **Recommend gradually integrating CycleResearcher into existing research workflows**
>    - Start with literature reviews and hypothesis generation, expanding to experimental design as confidence in the tool grows
>    - **Maintain regular human oversight and discussion of model suggestions, treating it as a sophisticated research assistant rather than an autonomous researcher**
>    - Document all model-generated suggestions and clearly distinguish between AI-assisted and human-guided portions in research
>
> These guidelines aim to fully utilize CycleResearcher's capabilities while maintaining scientific rigor and research integrity. **We emphasize that the model should be used as a tool to enhance human research capabilities, not as a substitute for human scientific judgment**. By following these guidelines, researchers can better utilize AI tools to improve research efficiency and quality while ensuring research process controllability and reliability.
>
> Best regards,
>
> Authors of “CycleResearcher: Improving Automated Research via Automated Review”
>
> ---
>
> [1] Locking Down the Finetuned LLM Safety
>
> [2] Fine-tuning Aligned Language Models Compromises Safety, Even When Users Do Not Intend To!
>
> [3] Fast-DetectGPT: Efficient Zero-Shot Detection of Machine-Generated Text via Conditional Probability Curvature

---

### Meta-Review · Area_Chair_PfL8 · 2024-12-23

**Metareview:**

The authors study the problem of automatically evaluating and writing scientific papers. They introduce 1) CycleReviewer, a model that takes in a paper and returns a score from 1-10 (trained to match the score a human peer reviewer would give), and 2) CycleResearcher, an automated paper writing model that trains against the reward provided by CycleReviewer.

This was an interesting but contentious paper. Reviewers all found the work intriguing, but raised questions about the ethical implications of the work, the overclaiming in the writing, as well as the rigor of the evaluations. The authors significantly revised their manuscript and experiments in response. These responses partially addressed the reviewers' concerns, but not fully. I thank the authors and reviewers for their efforts during the rebuttal phase.

After discussion, I recommend acceptance but with reservations. I **strongly** encourage the reviewers to incorporate the following feedback into their camera-ready version and to consider the ethics of releasing their model publicly.


1. The main claims of the paper -- that CycleResearcher produces papers that are comparable in quality to human-authored papers -- rely mainly on CycleReviewer scores. CycleReviewer is trained from historical data to mimic human reviewer scores. However, this historical data wholly consists of (one would hope) human-written papers, which are significantly out-of-distribution to the generated papers. For example, the initial version of this manuscript used LLMs to autoregressively generate all of the experimental results (which, as the reviewers pointed out, involves a lot of hallucination). Nevertheless, CycleReviewer gave those papers high scores. In human peer review, there is a certain benefit of the doubt given that the authors actually correctly did what they said they did; reviewers do not often, for instance, read through every line of the submitted code to ensure that it is correct. For generated papers, a much higher standard of scrutiny needs to be applied. Furthermore, since CycleResearcher is trained directly to optimize CycleReviewer scores, it is also not surprising that it does well by the metric of CycleReviewer scores.

    In response to similar reviewer comments, the authors (a) prompted OpenAI o1 to generate and then execute code, instead of just making up experimental results; and (b) ran a n=3 human reviewer evaluation study. While (a) seems like a good step towards more reliable results, it still needs to be validated. There is no analysis of the code, and its correctness, provided in the paper. Likewise, there are few details provided on (b) -- for example, was the "author" of the papers blind to the reviewers? Were the reviewers able to pick up errors in the experiments/results? What validation was done to show that the human study was reliable? Both (a) and (b) need to be elaborated upon and made much stronger to convince readers that CycleResearcher can actually do end-to-end research.


2. I recognize that running a rigorous evaluation of the generated papers is difficult and could be the topic of a different project. Without introducing more experiments, I strongly recommend that the authors shift the framing of their paper to more accurately reflect their experiments. For example, these results present an interesting example of how one can generate papers that optimize for automated review metrics without necessarily having the results in those papers be anywhere correct. The current, already-toned-down claim that CycleResearcher generates papers "with an average quality level close to human-written preprints" is not substantiated by the current experiments.

**Additional Comments On Reviewer Discussion:**

The points in the meta-review were all brought up by one or more reviewers. In response, the authors worked on (a) and (b) as summarized in the meta-review.

---

### Decision · Program_Chairs · 2025-01-22

Accept (Poster)